
**Earthquake Vulnerability Assessment of the Built Environment in Srinagar City, Kashmir Himalaya, Using GIS**

Midhat Fayaz[1], Shakil Ahmad Romshoo[1, 2, 3,#], Irfan Rashid[1] and Rakesh Chandra[2, 4]

[1]Department of Geoinformatics, University of Kashmir, Hazratbal Srinagar, 190006, Jammu and Kashmir, India

[2]Department of Earth Sciences, University of Kashmir, Hazratbal Srinagar, 190006, Jammu and Kashmir, India

[3]Islamic University of Science and Technology (IUST), Awantipora, 192122, Jammu and Kashmir, India

[4]Department of Geology, University of Ladakah, Leh

[#]*Correspondence to*: Shakil Ahmad Romshoo (shakilrom@kashmiruniversity.ac.in)

**ABSTRACT**

The study investigates the earthquake vulnerability of buildings in Srinagar, an urban city in the Kashmir Himalaya, India. The city, covering an area of around 246 km$^2$ and divided into 69 municipal wards, is situated in the tectonically active and densely populated mountain ecosystem. Given the haphazard development and high earthquake vulnerability of the city, it is critical to assess the vulnerability of the built environment to inform policymaking for developing effective earthquake risk reduction strategies. Integrating various parameters in GIS using Analytical Hierarchical Process (AHP) and Technique for Order Preference by Similarity to Ideal Solution (TOPSIS) approaches, the ward-wise vulnerability of buildings revealed that a total of ~17 km$^2$ area (~7% area; 23 wards) has very high to high Vulnerability; Moderate Vulnerability affects ~69 km$^2$ of the city area (28 %; 19 wards); ~160 km$^2$ area (~65% area; 27 wards) has vulnerability ranging from very low to low. Overall, the downtown city is most vulnerable to earthquake damage due to the high risk of pounding, high building density, and narrower roads, with little or no open spaces. The modern uptown city, on the other hand, has lower earthquake vulnerability due to the relatively wider roads and low building density. To build a safe and resilient city for its 1.5 million citizens, the knowledge generated in this study would inform action plans for developing earthquake risk reduction measures, which should include strict implementation of the building codes, retrofitting of the vulnerable buildings and creating a disaster consciousness among its citizenry.

Keywords: Earthquake, Earthquake Vulnerability, AHP, GIS, TOPSIS, Kashmir


## 1.    Introduction

Among all the natural disasters earthquakes are unique in the way that they occur without warning (Langenbach, 2009) and are a major hindrance in the way of achieving sustainable development. Cities are growing fast all over the world as a process of urbanization and more than half of the world's population lives in urban areas (Ritchie and Roser, 2018). Earthquakes cause immense loss of lives and damage to properties, livelihoods, economic infrastructures and communities, particularly in major urban centres (Kjekstad and Highland, 2009). Urban earthquake vulnerability has increased over the years due to the increasing complexities in urban built environments (Düzgün et al., 2009; Riedel et al., 2015). The high earthquake susceptibility of urban centres is also attributed to their situation in hazard-prone locations (Duzgun et al., 2011; Mir et al., 2017), haphazard urbanisation (Jena et al., 2020), and growing population (Beck et al., 2012) and has attracted the attention of emergency planners in estimating the seismic risk associated with future earthquakes (Kontoes et al., 2012). Surveys have shown that collapsing buildings and other physical structures during an earthquake cause huge social, economic and human losses (Panahi, et al., 2014). The dynamic interaction between different urban components and diverse forms of vulnerability proves that vulnerability is inherently a spatial problem (Hashemi and Alesheikh, 2012). This marks that the earthquake vulnerability of a building is an important parameter in the evaluation of earthquake potential damages in urban fabrics (Amini et al., 2009). Thus, assessment of earthquake vulnerability of the built environment is crucial for any city located in an earthquake risk zone to better understand the inherent weakness and vulnerabilities of the city against earthquakes and to help prioritize preparedness and risk mitigation activities.

Structural vulnerabilities to earthquakes have arisen in Kashmir in  recent decades when the traditional construction material and practices have been abandoned in favour of new practices (Yousuf et al., 2020). The lurking threat of an earthquake had in the past a great influence on the way people traditionally used to build their houses in the Kashmir valley (Langenbach, 2009; Ahmad et al., 2017). Traditional wood-frame structures were designed to deal with earthquake threats to provide a safe and suitable built environment for the people. The buildings built with wood substantially reduce the weight of buildings and provide structural flexibility compared to that of the other types of materials used in housing constructions (Alih and Vafaei, 2019). Recently, the traditional ways of constructing houses have been replaced mostly by concrete types, thereby increasing the vulnerability of the structures to earthquakes. It is therefore very important to assess the earthquake vulnerability



of all the existing buildings in the Kashmir valley, comprising of the traditional and modern
construction types, since the valley falls in Seismic Zones IV or V (Ali and Ali, 2020).
Despite the high vulnerability of the Kashmir valley to earthquakes, no initiative has been
taken by the government and scientific community to develop earthquake risk assessment
strategy of the valley that would have informed urban development planning to minimize the
damage in the eventuality of an earthquake as has been done in other vulnerable Himalayan
areas of the country like Delhi, Dehradun, Kolkata, etc (Pathak, 2008; Nath, et al., 2015;
Rautela et al., 2015; Sinha et al., 2016).
Many national and international studies have been conducted to estimate the physical
vulnerability of the built environment by applying various techniques, viz., MCDM (Multi-
Criteria Decision Making) AHP (Analytical Hierarchical Process), ANN (Artificial NeuraL
Networking) (Jena et al., 2020; Jena and Pradhan, 2020; Lee et al., 2019; Alizadeh et al.,
2018). Rashed and Weeks, 2003  studied the physical vulnerability parameters for the Tabriz
city of Iran that are major contributors in assessing the vulnerability of buildings like the age,
height of the buildings and earthquake intensity. Erden and Karaman, 2012  investigated the
impact of systemic vulnerability parameters, such as topography, distance to the epicentre,
soil classification, liquefaction, and fault/focal mechanism using AHP for earthquake
vulnerability assessment of the Kucukekmece region of Istanbul Turkey. Pathak, 2008 carried
out the earthquake vulnerability assessment of Guwahati city using Rapid visual screening
(RVS) by taking into account demand-capacity computation and structural / non-structural
damage grade indexing. Nath et al., 2015 used geotechnical, seismological and geological
data for assessing the seismic risk of Kolkata city. They used land use/land cover, population
density, building typology, age and height for earthquake vulnerability assessment. Sinha et
al., 2016 used Spatial Multi-Criteria Analysis and Ranking Tool (SMART) methodology and
classified the capital city of India, Delhi, as highly vulnerable to earthquake disaster using
different physical parameters like the number of stories, year-built range, area, occupancy
and construction type. The earthquake vulnerability of Nanital and Mussorie cities of the
Uttarakhand state, India was assessed by Rautela et al., 2015 employing the RVS
methodology. Ahmad et al., 2012 used experimental and analytical studies to investigate
Half-Dressed rubble stone (DS) masonry structures of the Himalayas using the shake table
method and fragility analysis of buildings. The study concluded that about 40% of buildings
can collapse in the eventuality of a large earthquake. The collapse percentage of buildings
can go as high as 80% if the epicentre of an earthquake is closer to the site. Baruah et al.,
2020  have assessed the seismic vulnerability of the mega-city Shillong in India using RVS





methodology by including parameters like building typology, local geology, geomorphology,
slope angle, and population suggesting that 60% of the city is falling under moderate to high
vulnerability zones. Jena et al., 2021 carried out the earthquake vulnerability of the Indian
subcontinent using the LSTM (Long Short-Term Memory) model and multi-criterion
analysis, which suggested that very-high vulnerable areas are situated towards northern and
eastern parts of India. The study, conducted at a coarse scale, classified Jammu and Kashmir,
of which the study area is a part, into a highly vulnerable state with a moderate to high
vulnerability index.
The present study addresses the knowledge gap through the assessment of high-
resolution earthquake vulnerability of built environment at the ward level in order to identify
the vulnerable areas of Srinagar city, a major rapidly growing and seismically vulnerable
urban centre in the Kashmir valley. Based on a literature review, expert opinion and analyses
of the available data, a set of six indicators, such as building geometry, density, height,
typology, pounding possibility and road network were selected in this study for assessing
earthquake vulnerability of the built-up environment in the city. The structural vulnerability
of Srinagar city, which is located in an earthquake-prone zone, will inform urban planning
and development strategies to create a safe and secure built environment with adequate green
and open spaces, as well as make the city sustainable, as envisioned under UNDP Sustainable
Development Goals (SDGs) 11 for sustainable cities and communities.
**2.    Srinagar city**
Srinagar city, spread over an area of 246 km$^2$, lies between 74º 43´ and 74 º 52´ E longitudes
and 34º 0´ and 34º 14´ N latitudes and is divided into 69 administrative wards (Fig. 1). The
city is situated at an elevation of 1713  m amsl along both the banks of the centrally flowing
Jhelum River, which is a major tributary of the Indus River System. The city of Srinagar,
home to around 1.5 million populations, is an economic hub, a seat of administration and an
important urban center in the Kashmir Himalaya (Parry et al., 2012). The population of the
city is projected to increase to 1.83 million by 2031 (Farooq and Muslim, 2014). The city is
susceptible to high seismic hazards due to its peculiar geological setting (Sana, 2018), urban
setting (Gupta et al., 2020), demographic profile and tectonic setting (Chandra et al., 2018).
The city is surrounded by Himalayan boundary faults, which are capable of generating
destructive earthquakes that are well documented in the historical archives and recent
instrumental records as well (Sana, 2018; Gupta et al., 2020). There is a formidable history of
earthquakes that have shaken Srinagar in the past millennium and have caused huge loss of





human life and property (Table 1) (Rajendran and Rajendran, 2005; Langenbach, 2007;
Bilham et al., 2010; Bilham, 2019; Yousuf et al., 2020). As Srinagar is an old and historic
city, most of the areas grew organically without following any physical plan or building
codes for the construction of its built infrastructure (Yousuf et al., 2020). Furthermore, there
is cultural and socio-economic inequality within the city, with lower-middle-income groups
residing in the densely populated downtown wards and upper-middle class and wealthy
people residing in the uptown wards of the city. In such a situation, assessing earthquake
ward-wise vulnerability of the built environment is very critical for prioritizing risk reduction
activities to reduce the earthquake vulnerability of the city (Mouroux et al., 2006; Mili et al.,

144  2018).

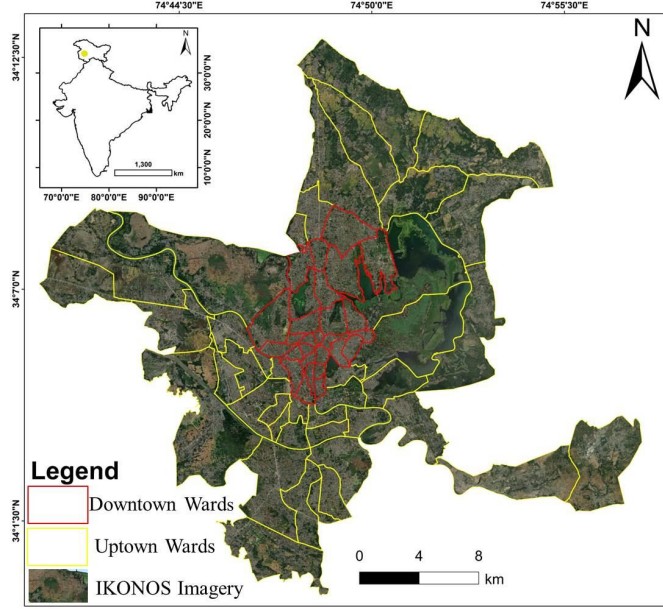


Fig. 1. Location of the study area. The ward boundaries are drapped on © Google Earth imagery
(IKONOS).

| S. No. | Date | Magnitude (M_w) | Lat (N) | Long (E) | Location | Damage | References |
|---|---|---|---|---|---|---|---|
| 1 | 844 AD | 6.5 to 7.5 | 34° N | 74.8° E | Srinagar Kashmir | Landslide dammed Jhelum at Khadanyar near Baramulla | Stein, 1982; Stein, 1898; Bilham and Bali, 2014. |
| 2 | 1123 AD | 6.5 to 7.5 | 34° N | 74.8° E | Srinagar Kashmir | Caused damage Sugandhesa Temple | Stein, 1982; Stein, 1898; Bilham and Bali, 2014; Iyengar |




| | | | | | | | and Sharma, 1996 ; Iyengar et al., 1999; Iyengar and Sharma, 1999 |
|---|---|---|---|---|---|---|---|
| 3 | 1501 AD | 6.5 to 7 | 34° N | 74.8° E | Srinagar Kashmir | Three months of after shocks | Bilham and Bali, 2014 |
| 4 | 1555 AD | 7.6 to 8 | 34.2 5° N | 74.8° E | Baramulla, Srinagar and Anantnag | Landslide, Liquefaction and landslides in the Kashmir valley | Bilham and Bali, 2014; Iyengar and Sharma, 1996 ; Iyengar et al., 1999; Iyengar and Sharma, 1999; Ambraseys and Jackson, 2003 |
| 5 | 1669 AD | 6.5 to 7 | 34° N | 74.8° E | Srinagar Kashmir | Mild shaking of buildings with no loss of life | Ahmad et al., 2009; Bilham and Bali, 2014 |
| 6 | 1678 AD | 6.5 to 6.8 | 34° N | 74.8° E | Kashmir | Continuous shaking of buildings | Ahmad et al., 2009; Bilham and Bali, 2014 |
| 7 | 1683 AD | 6.5 to 6.8 | 34° N | 74.8° E | Srinagar Kashmir | Long shocks and destruction of newly constructed houses | Ahmad et al., 2009; Bilham and Bali, 2014 |
| 8 | 1736 AD | 6.5 to 7 | 34° N | 74.8° E | Srinagar Kashmir | Large number of Building in city and adjoin areas collapsed completely | Ahmad et al., 2009 |
| 9 | 1779 AD | 6.5 to 7.5 | 34° N | 74.8° E | Srinagar and villages of Kashmir valley | It destroyed houses in city and villages and caused huge loss to life | Ahmad et al., 2009 |
| 10 | 1784 AD | 6.5 to 7.5 | 34° N | 74.8° E | Srinagar Kashmir | Terrific shocks felt in the area | Bilham, 2019 |
| 11 | 1828 AD | 6.5 to 7.5 | 34° N | 74.8° E | Srinagar Kashmir | About 1200 houses collapsed in this event | Vigne, 1842; Ahmad et al., 2009 |
| 12 | 1885 AD | 7.1 to 7.5 | 34.5 4° N | 74.68° E | Baramulla Kashmir | Terrific shock felt in the adjoining area | Ahmad et al., 2009; Lawrence, 1895 |
| 13 | 2005 AD | 7.6 | 34.4 9° N | 73.63° E | Kashmir | Earthquake alone left 86,000 people dead, about 69000 injured in both Indian and Pakistan side and about 25% of buildings were fully damaged in Uri and Poonch areas of J and K | Kumar et al., 2006 |

Table 1: Record of the past earthquake events in the Kashmir valley
**3.    Dataset and Methodology**
The availability of high spatial resolution satellite images with a ground pixel size of 1 m,
opens new possibilities for mapping individual features such as buildings (Li et al., 2019). To
accomplish this study, ortho-rectified Cartosat-2 data of 2016-17, having a spatial resolution
of 1 m, were utilised to extract the spatial information of the built environment in Srinagar
city.  The very high-resolution Cartosat-2 data has the potential to map individual buildings at
a large scale (Sandhu et al., 2021).




### 3.1 Building inventory

Keeping in view the advantages of manual delineation over digital image processing, the visual interpretation method was employed for delineating buildings and associated land use and land cover (Rashid et al., 2017). The image interpretation elements viz., tone, texture, pattern, size, shape, etc., supplemented by Google earth, was used to map the building footprint of the city on high-resolution Cartosat-2 data at a scale of 1:1000. All the buildings, roads, water bodies, and other associated urban built-up are included in the mapped features. Individual building footprints were accurately mapped, however delineating the complex geometrical shape in unplanned dense and very dense built-up areas proved to be a difficult task (Sandhu et al., 2021). As a result, rather than individual building footprints, building blocks were digitized in the densely populated areas towards the centre of the city where the edges of buildings become indistinguishable causing difficulty in extracting individual building footprints. Following evaluation in the field, these structures were segregated and corrected. Furthermore, all of the city's major roads were easily identifiable, however, the extraction of minor roads particularly in the dense built-up wards was difficult to map due to their narrower widths and metallic rooftop canopy of the adjacent building concealing the narrow alleys. The vector layer with the associated attribute like height, building occupancy, typology, and no. of floors database was created by combing remote sensing data and field data. The high-resolution building footprint and road network map were then utilized to critically assess the ward-wise earthquake vulnerability of buildings in the city.

### 3.2 Building vulnerability indicators

The vulnerability of the built environment determines its earthquake risk. Most of the damage during an earthquake is caused by building collapse. Thus faulty building structures and the use of unsafe material are some of the major causes of damage during an earthquake (Lantada et al., 2009). Assessment of earthquake vulnerability of individual buildings and neighborhoods is a complex process (Langenbach, 2009; Agrawal and Chourasia, 2007) and involves consideration of numerous parameters which are described as follows:

**3.2.1 Building height:** Because of its antiquity, tradition, heritage and significance, the built environment of different wards of Srinagar city shows a remarkable diversity (Meier and Will, 2008). Building height has a substantial impact on earthquake response and the level of structural damage (Kircher et al., 1997; Priestley, 2000). Buildings having a lower height-to-surface area ratio are more earthquake-resistant, and vice-versa (Alizadeh et al.,





2018). As a result, high-rise buildings with smaller surface area are more vulnerable to
earthquake damage. When these types of buildings shake and swing during an earthquake,
they have a higher probability of pounding. Extensive ward-by-ward field surveys were
conducted to generate a comprehensive building height map of the Srinagar city. During the
field surveys, the number of floors in randomly selected buildings from each ward in the city
was surveyed and counted. For height estimation during the field surveys, three types of
buildings were considered: single-story, double-story and triple or multiple storied buildings.
This field data was then combined in the GIS database.
**3.2.2    Masonry building:** Traditional construction practices are considered
outmoded, insubstantial and indicative of poverty in developing towns (Langenbach, 2009).
As a result, people are moving away from traditional types and methods of construction and
adopting modern practices and types of buildings with bricks, cement blocks and/or stones.
Masonry buildings, as they are known, are extremely vulnerable to earthquakes (Alam, and
Haque, 2018). The disappearance of traditional construction and buildings in Srinagar and the
rise of contemporary masonry construction practices make the city more vulnerable to
earthquakes.   A physical survey of buildings was conducted to determine the type of
buildings for physical vulnerability assessment of masonry buildings in the Srinagar city
(Rahman et al., 2015). The pattern of buildings along the main roads and link roads was
surveyed during the fieldwork because a majority of the buildings in the city are masonry.
The presence of building types other than masonry was recorded using Trimble Juno 5B
handheld GPS having 2 to 4 meter accuracy, which was then combined with GIS data to
estimate the proportion of various masonary building types in the city.
**3.2.3    Pounding Possibility:** One of the most common causes of structural damage
during an earthquake is pounding between neighboring buildings (Anagnostopoulos, 1988).
Pounding conditions occur when two or more buildings collide during an earthquake with a
smaller distance between them (Alam, and Haque, 2018). Every building has its natural
frequency and swings correspondingly during an earthquake (Lu et al., 2017; Jia et al., 2018).
If the separation distance between the buildings is insufficient, the buildings cannot swing
freely, resulting in local thrashing of the structures (Gioncu and Mazzolani, 2010). Due to the
location of the city in a seismically active region, socioeconomic setup, unplanned
urbanisation and faulty land-use planning (Yousuf et al., 2020), Srinagar city faces a
significant risk of structural damage from pounding during an earthquake. To determine the





potential of pounding in Srinagar, we employed a methodology that requires a minimum
separation distance between two buildings of 4% of the building height (FEMA, 1998). The
pounding potential was calculated using the following equation:
$$S = 0.04(h_1 + h_2) \qquad (1)$$
Where, 'S' is the minimum separation distance between the buildings, '$h_1$' and '$h_2$'
are the heights of two adjacent buildings.
**3.2.4   Building Geometry:** The earthquake damage to a building also depends on its
geometry. Compared to the regular structures, buildings having geometrical irregularities
such as a big height-to-width ratio, a large length-to-width ratio or a large offset in plan and
elevation perform poorly and sustain significant damage during earthquakes (Alih and
Vafaei, 2019). We employed high-resolution Cartosat-2 data and validated it against the field
data to generate a building geometry map of the city. The remote sensing data were pre-
processed and the edge enhancement technique was used for highlighting the edges of
buildings (Somvanshi et al., 2018; Huang et al., 2019). The geometry map of the city was
then generated using manual digitization of the building edges, which was later validated in
the field.
**3.2.5   Road Network:** Urban roadways are a complex network that is extremely
vulnerable to disruption in the event of natural disasters such as earthquakes (Golla et al.,
2020). Roads play an important role in the post-earthquake response and recovery phase.
Roadblocks caused by earthquakes have a negative impact on not just post-earthquake
emergency services but also isolate specific areas of cities where basic amenities such as
hospitals, shelters and other critical services are situated (Balijepalli and Oppong, 2014).
Thus, mapping of roads is essential for assessing the vulnerability of a city. Using a manual
digitization technique on the high-resolution satellite data, all roads of the Srinagar city
were mapped at a scale of 1:1000. Because the buildings in Srinagar are not built in a
planned manner, the majority of the roads are small and narrower and are classified into
three categories: less than 8 feet, 8 to 16 feet and more than 16 feet roads (Alam, and
Haque, 2018). Roads with a width of less than 8 feet are considered particularly vulnerable.
**3.2.6   Building density:** In addition to the aforementioned parameters, the building
density of an urban area has a significant impact on its structural vulnerability (Bahadori et
al., 2017). The more densely built a place is, the more vulnerable it is to earthquakes (Jena
and Pradhan, 2020). For all of the wards of Srinagar, the building density was determined as



the number of buildings per unit area. For building density mapping, we used 1m high-
resolution Cartosat data, which we then draped onto Google Earth imagery for validation.
The building density was also validated during the field surveys.
**3.3 Field validation**
Comprehensive ground-truth surveys were conducted in all wards throughout the city to
validate the building inventory database. Because there are so many buildings and their area
is so large, ward-wise validation of the mapped buildings was done using a stratified random
sampling method. It was ensured that the validation sites are well distributed throughout a
ward (Han et al., 2020). For field data collection, a proforma was developed to collect data,
such as latitude, longitude, building use, number of floors and construction type. The position
of individual buildings in every ward was identified on the building inventory map during
field surveys through visual observation and GPS coordinates and the locations were
documented (Ahmad et al., 2009). 8000 field validation points were collected throughout the
city (Fig. 2) and the physical attributes of each building were inspected externally to
determine building parameters such as building height, number of floors and type of
construction. Post-field surveys, the building inventory database was updated to match the
ground-truth data.

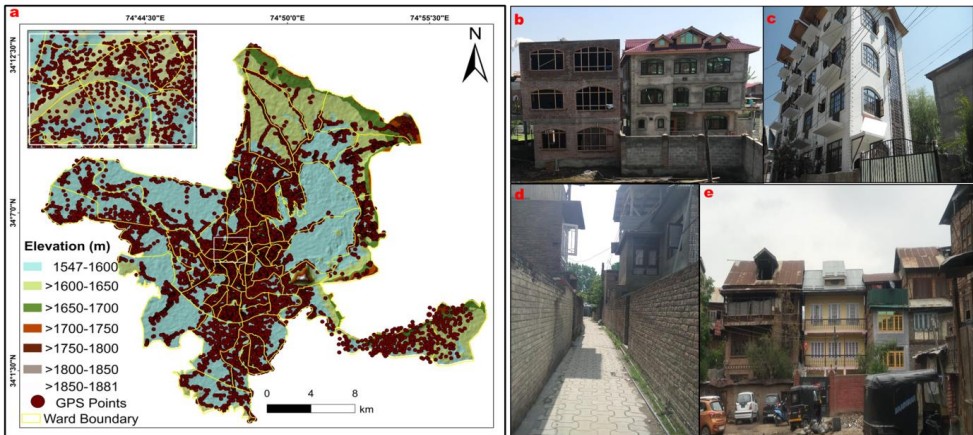

Fig. 2. *a)* Field validation map showing the distribution of ground samples with the inset
showing the density of samples. The elevation of study area are based on ASTER DEM data.
*Field photograph of b)* modern masonry construction practice adopted in residential *c)*
commercial building with large windowpanes *d)* narrower roads in the city centre *e)*
buildings with the insufficient or no separation distance.



### 3.4 Analytical Hierarchical Process (AHP) Approach

Due to its simplicity and rationality (Rezaie and Panahi, 2015; Alam and Mondal, 2019), the AHP is a widely used multi-criteria decision-making method (MCDM) for vulnerability assessment. It considers both qualitative and quantitative parameters to develop a hierarchical solution decision-making among various alternatives and their sub-categories. The Analytical Hierarchical Process (AHP) weights parameters and sub-parameters based on expert opinion, ensuring transparency and consideration of local specific conditions of a study area that global indices cannot (Füssel, 2010). There are three key assessment steps in AHP. The first step is to create binary comparison matrices on a scale of 1–9 (Saaty, 1980), where 1 indicates that two parameters are equally important, 9 indicates that one parameter is extremely important and 1/9 indicates that the parameter is of the least importance. Table 2 displays the scale of importance. The AHP was used to create indices that measured spatial variations in structural vulnerability ward-by-ward across the Srinagar city.

| Decreasing Relative Intensity of Importance | Equally Important | Increasing Relative Intensity of Importance |
|---|---|---|
| 1/9  1/8  1/7  1/6  1/5  1/4  1/3  1/2 | 1 | 2  3  4  5  6  7  8  9 |

Table 2: AHP scale used in this study

In the second step, the weights of different factors are determined from row-multiplied value (RMV), in un-normalized and normalized values using Equations (2) and (3).

$$\text{Unnormalized value, } mi = \sqrt[n]{RMV} \qquad (2)$$

$$\text{Normalized value} = \frac{mi}{\sum_{i=1}^{n} mi} \qquad (3)$$

Where, *mi* refers to the un-normalized value of the *i-th* parameter and n represents the total influential parameters.

The third and most important step of this model is to compute the consistency between judgements and weights. The consistency is calculated from the consistency index and consistency ratio employing equations (4) and (5). If the consistency ratio is <0.1, the pairwise comparison matrix is consistent and if it is >0.1, the pairwise comparison between indicators and sub-indicators must be iterated until a good consistency is achieved.

$$\text{Consistency index, } CI = \frac{L-n}{n-1} \qquad (4)$$

$$\text{Consistency Ratio, } CR = \frac{CI}{RI} \qquad (5)$$





Where, L represents the Eigen-value of the pairwise comparison matrix and RI is the
random inconsistency index which depends on the number of vulnerability assessment
parameters (n) used in the assessment. The variation of RI values for a different number of
parameters is shown in Table 3.

| N | 1 | 2 | 3 | 4 | 5 | 6 | 7 | 8 | 9 | 10 | 11 | 12 |
|---|---|---|---|---|---|---|---|---|---|----|----|----|
| RI | 0 | 0 | 0.58 | 0.90 | 1.12 | 1.24 | 1.32 | 1.41 | 1.45 | 1.49 | 1.52 | 1.54 |

Table 3: Random inconsistency indices (RI) for n = 1, 2. . . 12. (After Saaty, 1980)
Based on multiple expert judgments, a comparison matrix of six earthquake
vulnerability factors was established in this study (Yariyan et al., 2020). The geometric mean
of expert opinions was then calculated to compile all of the opinions into a single matrix
(Table 4). As a result, the factors are weighted and ranked on a scale of 0 to 1. The
Consistency Ratio (CR) of 1.24 was achieved, which indicates consistency in the pairwise
comparison of vulnerability factors (Saaty, 1980 ).

| Parameters | Average Floor Height | Masonry Building (%) | Pounding Possibility (%) | Irregular Building (%) | Average Road Width | Building Density | Sum | Weight |
|---|---|---|---|---|---|---|---|---|
| Average Floor Height (m) | 0.12 | 0.09 | 0.12 | 0.08 | 0.16 | 0.15 | 0.72 | 0.12 |
| Masonry Building (%) | 0.28 | 0.23 | 0.22 | 0.23 | 0.22 | 0.22 | 1.40 | 0.23 |
| Pounding Possibility (%) | 0.28 | 0.32 | 0.31 | 0.31 | 0.27 | 0.31 | 1.80 | 0.30 |
| Irregular Building (%) | 0.12 | 0.08 | 0.08 | 0.08 | 0.05 | 0.08 | 0.48 | 0.08 |
| Road Width (ft) | 0.08 | 0.11 | 0.12 | 0.15 | 0.11 | 0.09 | 0.67 | 0.11 |
| Building Density (per Ha) | 0.12 | 0.16 | 0.15 | 0.15 | 0.19 | 0.15 | 0.92 | 0.15 |

Table 4: Pair-wise matrix showing weights for each of the factors used in the AHP model



**3.5    Technique for Order Preference by Similarity to Ideal Solution (TOPSIS)**
**Approach**
The TOPSIS is a multi-criteria decision-making (MCDM) method that chooses alternatives
based on the distance between positive and negative ideal points (Hwang et al., 1993; Joshi
and Kumar,  2014). The TOPSIS model is based on the concept that the chosen alternative
should be the closest to the ideal solution while being the farthest from the negative ideal
solution. The important steps involved in the TOPSIS approach are listed below.
Step 1: Construction of normalized decision matrix using Equation (6)
$$\text{Normalize score}, r_{ij} = x_{ij}/\left(\sum x_y^2\right) \qquad (6)$$
Where, $x_{ij}$ is the score of option $i$ with respect to criterion $j$.
Step 2: Construction of weighted normalized decision matrix using Equation (7)
$$v_{ij} = w_j * r_{ij} \qquad (7)$$
where, $w_j$ is the weight for each criterion.
Step 3: Identifying the positive and negative ideal solutions. The positive (A⁺) and the
negative (A') ideal solutions are defined according to the weighted decision matrix using
equations (8) and (9) respectively
$$A^+ = \{V_1^+, V_2^+, \dots, V_n^+\}$$
$$\text{Where, } V_j^+ = \{\max(V_{ij}) if j \in J; \min(V_{ij}) if j \in J'\} \qquad (8)$$
$$A' = \{V_1', V_2', \dots, V_n'\}$$
$$\text{Where, } V_j' = \{\min(V_{ij}) if j \in J; \max(V_{ij}) if j \in J'\} \qquad (9)$$

Step 4: Calculating the separation distance of each alternative from the positive and negative
ideal solution using equations (10) and (11) respectively.
$$S_i^+ = \sqrt{\sum_{j=1} n(V_j^+ - V_{ij})^2} \qquad (10)$$




$$S_i^- = \sqrt{\sum_{j=1} n(V_j^{'} - V_{ij})^2} \qquad (11)$$

Where, $S_i^+$ is the distance from the $i$[th] alternative from the positive ideal point for the
$j$[th] feature and $S_i^-$ is the distance between the $i$[th] alternative and the negative ideal point for
the $j$[th] feature and $i = 1, ..., m$. The negative and positive ideal point for each seismic
vulnerability factor is shown in Table 5.

| Vulnerability Parameters | Positive Ideal Point (V+) | Negative Ideal Point (V-) |
|---|---|---|
| Average Floor Height | 0.0171 | 0.0112 |
| Pounding Possibility | 0.0501 | 0.0083 |
| Irregular Building | 0.0270 | 0.0004 |
| Road Width | 0.0090 | 0.0199 |
| Building Density | 0.0618 | 0.0007 |
| Masonry Building | 0.0283 | 0.0243 |

Table 5: Positive and negative ideal points used in the TOPSIS model
Step 5: Measuring the relative closeness of each parameter to the ideal solution using
Equation (12).
$$\text{Closeness}, C_i^* = S_i^- / (S_i^- + S_i^+) \qquad (12)$$

Where, $C_i^*$ is a value between 0 and 1 and the closer the number is to 1, the closer the
alternative is to the ideal condition. The positive ideal point in this study is the one with the
maximum structural earthquake vulnerability, while the negative ideal point is the one with
the lowest vulnerability. Furthermore, the closer an alternative value is to 1, the more
vulnerable those limits are, and the closer it is to 0, the less vulnerable they are.
Based on expert opinions, the AHP model was used to assign weights to all the
parameters. Following that, the TOPSIS model was used to rank the wards after evaluating
the best alternatives using mathematical calculations. Finally, the weighted and best
alternative evaluated structural vulnerability parameters from both the AHP and TOPSIS
models are combined in the GIS environment to create a ward-by-ward earthquake
vulnerability map of the built environment for Srinagar.
**4.     Results and discussion**
**4.1. Analysis of building parameters:**
**4.1.1. Building height:** In the city, around 2.5 lakh buildings were mapped (Fig. 3),
with nearly 86.4% of the buildings being residential, 7.1 % being commercial, and the





remaining ~6.5% having various uses and purposes such as educational, religious, defence,
health and medical, industrial etc. The analysis revealed that single storey buildings account
for ~8% of all buildings, double-storey buildings account for ~50% and triple-story buildings
account for ~42%. However, only a small number (n=307, 0.12%) of buildings have more
than 3 floors. 18 of the 69 wards have an average of two floors while 51 have an average of
three floors.

The building height has a significant impact on the earthquake vulnerability of a ward.
A majority of the residential buildings in Srinagar have an average floor height of three
meters whereas; government offices and commercial buildings typically have an average
floor height of 3.5 meters. The lowest ward-wise average building height of 6.33 meters was
found in the municipal ward A (BB Cant), which is primarily a cantonment area used and
administered for security and defence purposes. Ward number 50 (Lal Bazar) in Srinagar has
the highest ward-wise average building height of 9.68 meters. Figure 4 depicts the spatial
distribution of ward-wise average building heights with the average values provided in Table
6.

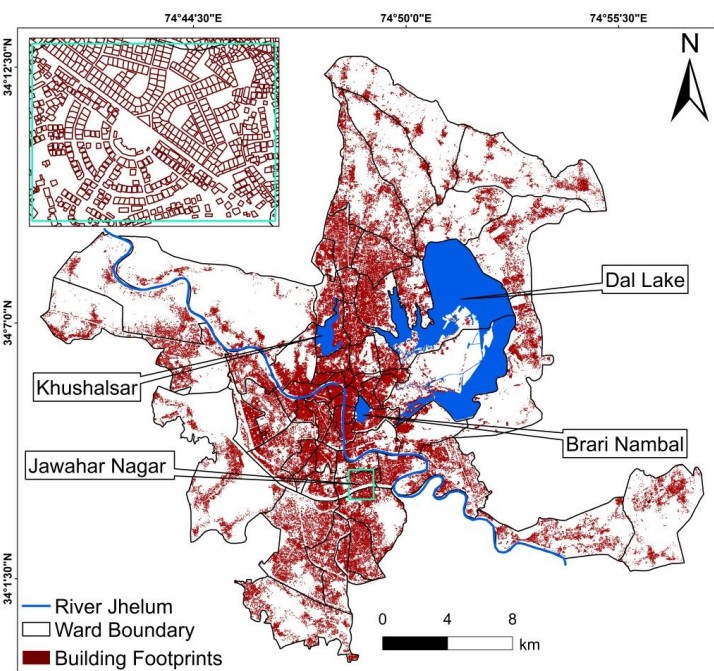


Fig. 3. Building foot print map of Srinagar city.





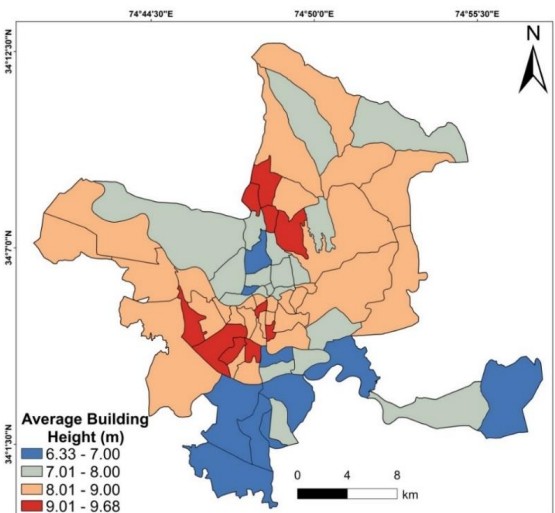


Fig. 4. Ward-wise distribution of average building height in Srinagar city.

| Ward No. | Ward Names | Irregular Buildings (%) | Pounding Possibility (%) | Masonry Buildings (%) | Building Density (per Ha) | Average Height (m) | Average Plinth Area (m²) | Average Road Width (ft) | Road Density (km/km²) |
|---|---|---|---|---|---|---|---|---|---|
| A | BB Cant | 4.01 | 40.58 | 98.64 | 3.86 | 6.33 | 149.41 | 9.61 | 6.23 |
| 1 | Harwan | 4.81 | 76.71 | 96.57 | 2.82 | 8.86 | 140.19 | 8.78 | 7.55 |
| 2 | Nishat | 3.07 | 56.33 | 98.17 | 2.34 | 8.64 | 124.19 | 8.16 | 6.08 |
| 3 | Dalgate | 2.01 | 36.85 | 85.98 | 3.76 | 7.70 | 128.59 | 9.85 | 7.95 |
| 4 | Lalchowk | 6.50 | 81.07 | 90.06 | 4.06 | 8.24 | 141.67 | 12.14 | 11.61 |
| 5 | Rajbagh | 3.17 | 46.41 | 97.46 | 7.59 | 7.37 | 130.12 | 8.78 | 15.47 |
| 6 | Jawahar Nagar | 6.08 | 73.48 | 98.41 | 6.92 | 7.39 | 182.51 | 11.20 | 19.77 |
| 7 | Wazir Bagh | 8.76 | 85.58 | 92.29 | 6.05 | 6.64 | 163.73 | 12.02 | 13.93 |
| 8 | Mehjoor Nagar | 0.95 | 60.43 | 99.75 | 7.30 | 6.88 | 115.25 | 8.79 | 12.38 |
| 9 | Natipora | 2.21 | 59.55 | 99.48 | 9.12 | 7.00 | 138.15 | 10.12 | 19.53 |
| 10 | Chanapora | 3.25 | 72.44 | 99.61 | 8.71 | 6.79 | 121.89 | 11.33 | 23.71 |
| 11 | Bagat-I-Barzullah | 3.80 | 46.86 | 99.32 | 4.87 | 6.90 | 152.71 | 10.09 | 13.99 |
| 12 | Rawalpora | 6.03 | 53.65 | 98.66 | 5.37 | 6.92 | 161.29 | 10.88 | 17.20 |
| 13 | Sheikh Dawood Colony | 1.32 | 55.38 | 97.23 | 9.73 | 8.39 | 129.31 | 7.55 | 14.97 |
| 14 | Batamaloo | 2.95 | 84.64 | 96.97 | 11.05 | 9.41 | 158.01 | 7.94 | 19.85 |
| 15 | Aloochi Bagh | 1.88 | 69.81 | 99.36 | 6.72 | 8.15 | 130.03 | 8.78 | 14.53 |
| 16 | Magarmal Bagh | 3.48 | 74.59 | 97.15 | 11.11 | 9.33 | 120.35 | 9.07 | 18.32 |
| 17 | Nund Reshi Colony | 3.24 | 79.26 | 97.47 | 4.99 | 9.05 | 184.49 | 10.08 | 12.21 |
| 18 | Qamarwari | 0.90 | 49.97 | 96.33 | 11.43 | 8.44 | 98.93 | 8.96 | 19.24 |
| 19 | Parimpora | 2.78 | 52.64 | 96.45 | 6.66 | 8.06 | 114.43 | 8.82 | 14.53 |




| 20 | Zainakote | 3.00 | 34.94 | 95.34 | 3.16 | 8.11 | 152.05 | 9.14 | 9.46 |
| 21 | Bemina East | 3.00 | 67.03 | 94.70 | 6.19 | 8.97 | 147.59 | 12.64 | 17.17 |
| 22 | Bemina West | 2.42 | 89.56 | 96.87 | 7.45 | 9.64 | 143.21 | 13.60 | 19.60 |
| 23 | Shaheed Gunj | 3.36 | 85.64 | 97.33 | 11.00 | 8.76 | 95.20 | 12.06 | 24.37 |
| 24 | Karan Nagar | 3.81 | 72.94 | 96.78 | 11.83 | 8.31 | 125.08 | 13.57 | 26.42 |
| 25 | Chattabal | 0.83 | 69.54 | 98.18 | 18.38 | 8.30 | 100.54 | 8.08 | 26.83 |
| 26 | Syed Ali Akbar | 0.61 | 87.41 | 87.41 | 24.53 | 8.50 | 87.57 | 6.12 | 35.38 |
| 27 | Nawab Bazar | 1.12 | 77.45 | 93.01 | 19.90 | 8.19 | 97.25 | 8.60 | 33.67 |
| 28 | Islamyarbal | 0.58 | 82.50 | 96.51 | 25.96 | 9.40 | 73.68 | 7.72 | 34.46 |
| 29 | Aali Kadal | 0.63 | 84.02 | 99.81 | 29.97 | 8.64 | 81.89 | 7.70 | 39.33 |
| 30 | Ganpatyar | 1.04 | 96.27 | 98.66 | 18.58 | 9.45 | 130.66 | 6.42 | 37.36 |
| 31 | Bana Mohalla | 0.54 | 72.28 | 99.76 | 21.75 | 8.79 | 103.71 | 6.14 | 40.22 |
| 32 | Sathoo Barbarshah | 1.46 | 77.23 | 95.05 | 9.21 | 8.05 | 121.44 | 8.69 | 16.15 |
| 33 | Khankai Moulla | 1.05 | 76.08 | 98.61 | 23.95 | 8.18 | 87.06 | 7.10 | 39.79 |
| 34 | S R Gunj | 1.56 | 91.10 | 99.69 | 22.65 | 7.86 | 86.02 | 7.20 | 42.29 |
| 35 | Aqilmir Khanyar | 1.52 | 93.14 | 99.68 | 22.14 | 8.05 | 94.67 | 8.40 | 28.82 |
| 36 | Khawaja Bazar | 1.60 | 97.11 | 99.77 | 24.90 | 7.82 | 73.60 | 9.55 | 27.80 |
| 37 | Safa Kadal | 2.90 | 80.12 | 99.36 | 16.43 | 7.82 | 113.22 | 6.86 | 27.27 |
| 38 | Iddgah | 2.05 | 88.71 | 99.53 | 8.38 | 7.74 | 110.69 | 9.19 | 13.15 |
| 39 | Tarbal | 0.56 | 98.27 | 99.65 | 38.89 | 6.96 | 71.17 | 7.91 | 38.42 |
| 40 | Jogi Lankar | 2.33 | 91.62 | 99.36 | 16.93 | 8.46 | 97.27 | 7.37 | 25.51 |
| 41 | Zindshah Sahib | 4.37 | 97.94 | 99.07 | 23.73 | 8.40 | 95.92 | 6.17 | 29.72 |
| 42 | Hasanabad | 2.98 | 89.68 | 99.78 | 9.79 | 7.85 | 112.33 | 7.55 | 20.44 |
| 43 | Jamia Masjid | 1.58 | 99.66 | 97.27 | 46.35 | 7.77 | 61.48 | 7.51 | 40.53 |
| 44 | Makhdoom Sahib | 2.06 | 86.04 | 99.04 | 8.60 | 7.78 | 104.74 | 8.73 | 19.49 |
| 45 | Kawdara | 1.16 | 85.52 | 99.23 | 16.67 | 7.02 | 105.54 | 7.74 | 26.03 |
| 46 | Zadibal | 0.87 | 42.34 | 99.69 | 7.20 | 6.96 | 108.13 | 10.11 | 12.63 |
| 47 | Madin Sahib | 2.57 | 70.85 | 99.59 | 13.82 | 7.46 | 103.58 | 11.75 | 22.59 |
| 48 | Nowshera | 3.86 | 65.95 | 99.56 | 8.59 | 9.56 | 145.24 | 8.29 | 20.11 |
| 49 | Zoonimar | 2.28 | 41.22 | 99.66 | 7.56 | 7.52 | 126.14 | 8.40 | 17.12 |
| 50 | Lal Bazar | 5.45 | 93.81 | 99.30 | 9.43 | 9.68 | 147.35 | 10.07 | 16.22 |
| 51 | Umer Colony | 6.65 | 82.78 | 99.77 | 7.86 | 8.54 | 175.91 | 9.32 | 15.49 |





| | | | | | | | | |
|---|---|---|---|---|---|---|---|---|
| 52 | Soura | 3.24 | 79.14 | 98.01 | 9.39 | 9.62 | 105.28 | 9.89 | 17.59 |
| 53 | Buchpora | 2.43 | 47.26 | 99.62 | 8.34 | 9.55 | 147.94 | 9.70 | 23.36 |
| 54 | Ahmad Nagar | 5.24 | 79.42 | 99.58 | 4.04 | 8.77 | 167.24 | 8.69 | 9.73 |
| 55 | Zakora | 3.29 | 63.88 | 99.67 | 2.02 | 7.38 | 154.04 | 8.98 | 5.92 |
| 56 | Hazratbal | 6.15 | 83.01 | 96.86 | 4.50 | 8.01 | 158.51 | 11.41 | 11.89 |
| 57 | Tailbal | 1.19 | 53.49 | 99.25 | 2.86 | 8.54 | 106.30 | 8.29 | 7.44 |
| 58 | Bud Dal | 0.73 | 58.98 | 98.74 | 0.49 | 8.86 | 82.14 | 6.14 | 1.35 |
| 59 | Locut Dal | 1.02 | 86.53 | 87.24 | 1.80 | 8.76 | 72.43 | 6.75 | 1.79 |
| 60 | New Theed | 0.86 | 46.17 | 99.08 | 2.03 | 7.84 | 108.99 | 7.89 | 5.63 |
| 61 | Alasteng | 2.42 | 71.43 | 99.25 | 1.74 | 8.02 | 126.34 | 8.17 | 3.93 |
| 62 | Palapora | 0.14 | 28.23 | 99.46 | 1.33 | 7.43 | 83.49 | 8.16 | 2.97 |
| 63 | Maloora | 0.75 | 24.50 | 98.09 | 1.56 | 8.40 | 146.59 | 9.52 | 5.56 |
| 64 | Lawaypora | 1.49 | 39.23 | 99.03 | 1.64 | 8.51 | 143.06 | 9.79 | 4.91 |
| 65 | Khumani Chowk | 1.00 | 90.18 | 99.57 | 1.79 | 8.26 | 112.34 | 7.74 | 4.85 |
| 66 | Humhama | 2.60 | 22.50 | 99.36 | 3.27 | 6.83 | 131.51 | 10.45 | 9.42 |
| 67 | Pantha Chowk | 2.63 | 18.02 | 99.21 | 2.77 | 7.01 | 105.17 | 9.51 | 5.91 |
| 68 | Khonmoh | 1.70 | 16.59 | 99.37 | 2.14 | 6.99 | 89.06 | 9.54 | 5.11 |

Table 6: Ward-wise built-up parameters used for vulnerability assessment of the Srinagar city.

**4.1.2 Masonry Building:** The type of construction material used in building construction determines the earthquake vulnerability of the built environment (Lang et al., 2018). The masonry buildings (those constructed of bricks, cement blocks or stone) have an extremely poor seismic performance (Alam, and Haque, 2018). The strength of the buildings is mostly determined by the materials used for the walls and the type of mortar used (Lang et al., 2018). Table 6 and Fig. 5 show the ward-wise distribution of masonry buildings in Srinagar. The proportion of masonry structures in the city varies between 82% and 99.8% in different wards. Masonary buildings account for about 98% of the city's total buildings making it highly vulnerable to earthquakes. Ward number 29 (Aali kadal) has the largest number of masonry buildings (99.8%), whereas, wards 3, 26 and 59 (Dalgate, Syed Ali Akbar, Jawahar Nagar, respectively have about 15% non-masonry buildings.

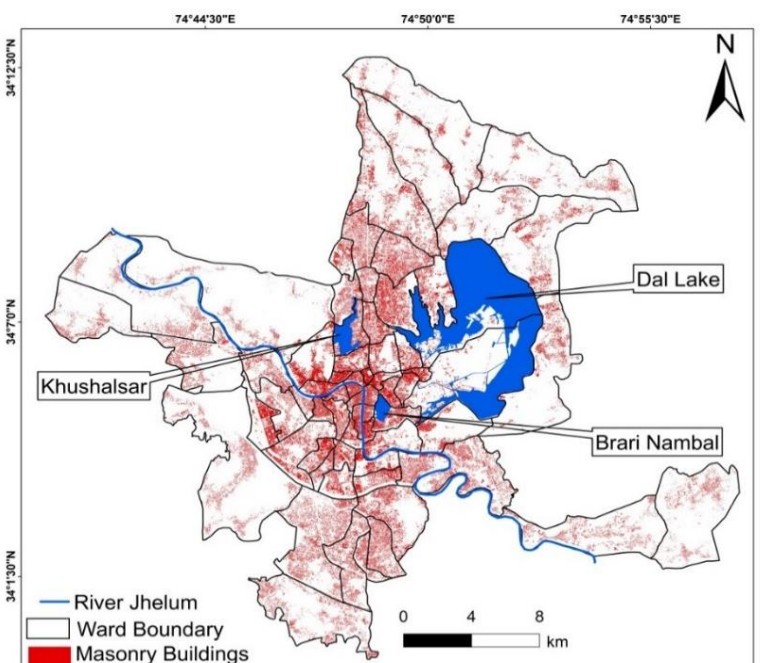


Fig. 5. Ward-wise distribution of Masonry buildings in the city.
**4.1.3    Pounding possibility:** From the analysis of the estimated separation distance
and height of adjacent buildings, it was found that ~ 65% of buildings in the city have the
high chance of pounding with neighboring buildings, at least on one side, because the ideal
offset between the buildings has not been maintained due to the haphazard building
construction practices, particularly in the downtown wards of the city (Fig. 6). Table 6
provides information about the ward-by-ward pounding probability of the city. It is therefore
evident from the analysis that the downtown wards of the city have the highest risk of
pounding because the buildings are densely packed in most of the wards. Comparably, the
uptown wards show a lower pounding possibility due to the sufficient gaps between the
buildings.


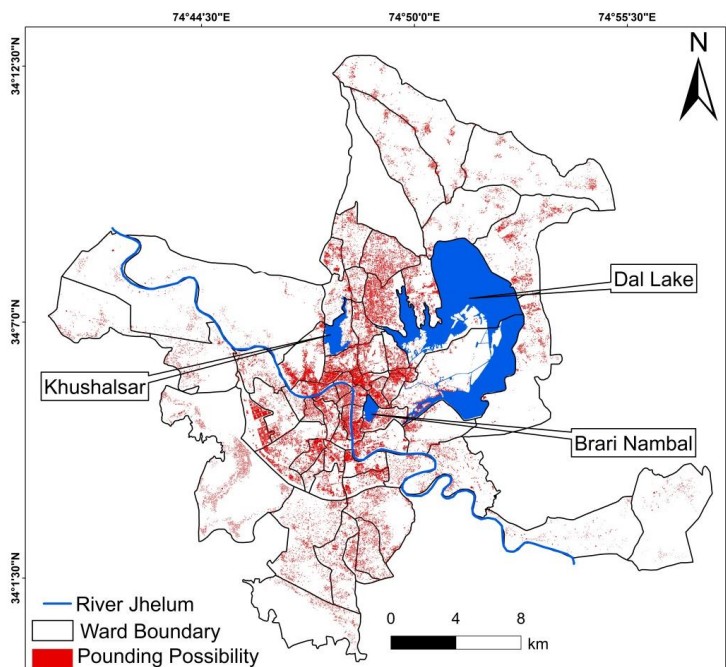


Fig. 6. Ward-wise distribution of building pounding possibility in the city.

**4.1.4 Building Geometry:** Modern buildings in the city are constructed with irregular shapes and frequent offsets for aesthetic building layout and structural design. The building irregularities either plain or vertical make the structures vulnerable to seismic loading (Mazza, 2014; Ahirwal et al., 2019). As a result, while assessing the earthquake vulnerability of the built environments, building irregularity is a very important factor to consider. It was found from the analysis of the data provided in Table 6 and Figure 7 that ~3% of the buildings in the city have irregular shapes. A fewer number of irregular buildings are found in the municipal ward number 62 i.e Palapora (n=8, 0.13%), whereas the largest number of irregular buildings are present in ward number 7 i.e Wazir Bagh (n=158, 8.76%), increasing the ward's vulnerability in the city. The typical residential buildings usually have a conventional regular and rectangular shape with four sides and an average plinth area of 120 m$^2$ (Table 6). Some of the schools, colleges, government offices, hospitals and commercial complexes have irregular architectural shapes, such as the shape of the letters "O", "L", "U", "T", and "H" making them more vulnerable to earthquakes. Furthermore, most schools, colleges and hospitals are usually made up of multiple smaller building units with regular




shapes and are close to each other increasing the risk of pounding and making these building
complexes more vulnerable to earthquake damage.

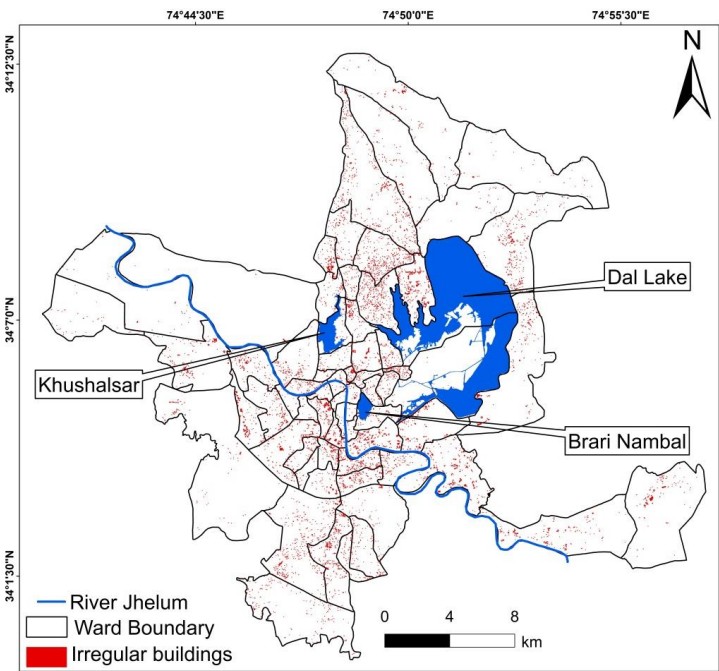


Fig. 7. Ward-wise distribution of irregular shaped buildings in the city.
**4.1.5   Building Density:** The average building density of Srinagar is ten buildings
per hectare (including residential and commercial buildings). However, the building density
in 17 wards of the downtown city is more than 15 buildings per hectare (Table 6; Fig. 8). The
highest building density of 46 buildings per hectare was observed in the municipal ward
number 43 (Jamia masjid), followed by the wards 39 (Tarabal) and 29 (Aali kadal), which
have a building density of 39 and 30 respectively. Ward number 58 (Bud Dal) has the lowest
building density, with only one building per hectare. Knowledge about the building packing
within the urban city centre is crucial information for the earthquake vulnerability
assessment. The current practice of constructing buildings with insufficient space between
them increases the congestion and building density of cities (Bahadori et al., 2017). The areas
with high building density (Table 6) are more vulnerable to earthquake damage than areas
with low building density (Shadmaan and Islam, 2021). The high building density also leads
to a small separation distance between buildings and a reduction in the open space area. This
reduces the amount of useful space available for evacuation and shelter during post-



earthquake rescue operations. In order to decrease the loss and damage to human life and
infrastructure caused by earthquakes, it is important to regulate building density and ensure
the reinforcement of old structures (Jena et al., 2020). Good planning, lower building density,
and evenly spaced buildings can reduce the seismic vulnerability of a city (Aghataher et al.,
2018).

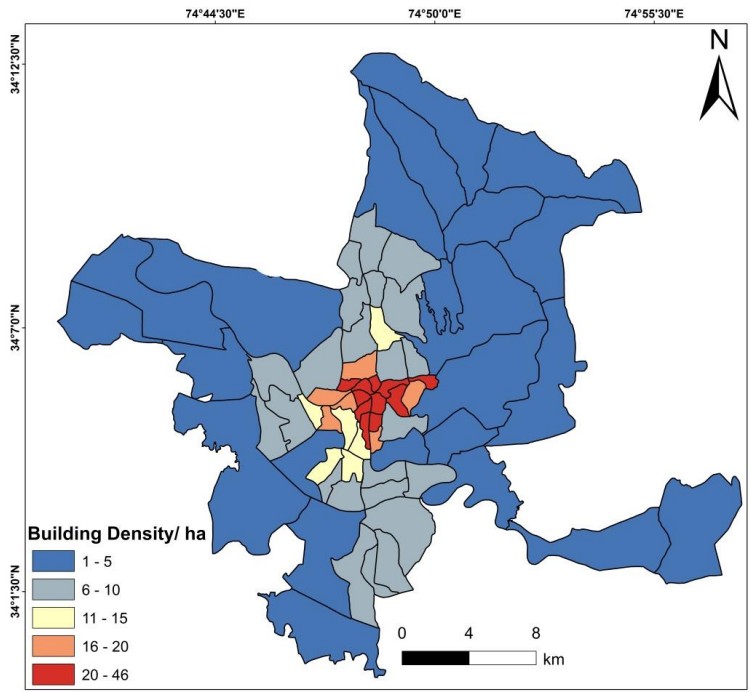


Fig. 8. Ward-wise distribution of building density in the city.

**4.1.6   Road Network:** Despite the high population and building density in the city,
the road network connectivity in the city is good with a total road length of 2246 kilometres.
In the eventuality of an earthquake, the effectiveness of the urban road network decreases
significantly due to road damage caused by the collapsed buildings and blockages (Bono and
Gutiérrez, 2011; Zanini et al., 2017). On the basis of the width, the roads were classified
into three categories viz., <8 ft, 8 to 16 ft and > 16 ft (Fig. 9). The roads or streets with a
width <8 ft are considered possible blockade sites. From the analysis of the data provided in
Table 6, it is evident that wards 26 (Syed Ali Akbar), 31 (Bana Mohalla), 58 (Bud Dal), 41
(Zind Shah Sahib), 30 (Ganpatyar), and 37 (Safa kadal) have the smallest average road width
of less than 7 ft., despite having high road density except for the ward 58(Bud Dal), which
has a road density is 1.35 km km$^{-2}$ due to the fact that most of the ward is covered by water
(Dal Lake). Ward 26 has a road density of 35.38 km km$^{-2}$, ward 31 has a road density of
40.22 km km$^{-2}$, ward 41 has a road density of 29.72 km km$^{-2}$, ward 30 has a road density of
37.36 km km$^{-2}$ and ward 37 has a road density of 27.27 km km$^{-2}$. Wards 24 (KaranNagar) and
22 (Bemina West) have the largest average road width of 13.58 ft with a road density of
26.42 and 19.60 km km$^{-2}$, respectively (Table 6). It is worth noting that the road network in
the city is relatively denser in the downtown city and as a result, the roads being narrower
makes these places in the city more vulnerable to earthquake damage and possibly impeding
the post-earthquake evacuation and rehabilitation operations. The road network in the

uptown wards towards the periphery of the city, on the other hand, is less dense. The

roads being relatively wider in the outer wards make them more suitable for evacuation and
would facilitate easy movement of traffic and relief during an earthquake compared to the
inner city wards.

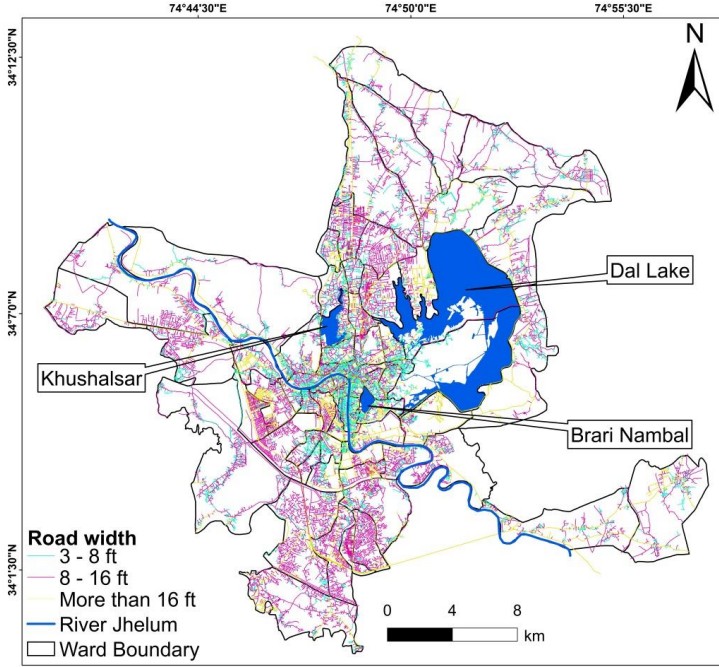

Fig. 9. Ward-wise road network in the city.
**4.2     Earthquake vulnerability Analysis:**
Earthquake events are common in the Kashmir valley and they are characterised by high
exposure to social and economic consequences that can be severe (Oliveira,   2003).
Earthquake vulnerability assessment aids pre-earthquake planning and post-earthquake
emergency operations by providing vital information that informs earthquake risk reduction
measures (Saputra et al., 2017). The GIS-based analysis of earthquake vulnerability of the
built environment in Srinagar, using the coupled model of AHP and TOPSIS was carried out
to highlight the ward-wise vulnerability in the event of an earthquake. Because all of the
structural vulnerability parameters have different importance and impact, the structural
vulnerability of the city cannot be achieved by relying on a single parameter (Panahi, et al.,
2014). Therefore, all of the important six parameters were considered in this study to produce
a good earthquake vulnerability assessment of the city.  This study classified 69 municipal
wards of the city into five earthquake vulnerability classes: very high, high, moderate, low
and very low earthquake vulnerability. The results showed that 9 municipal wards in the city
are very highly vulnerable, 14 wards are highly vulnerable, 19 wards are moderately
vulnerable, 17 wards are low vulnerable and 10 wards fall in the very low vulnerable
category (Fig. 10).  The vulnerability map reveals that wards categorized under the same
vulnerability class are contagious to one another, indicating a clear pattern of earthquake
vulnerability in Srinagar. The city centre, which also happens to be the site of ancient urban
settlements including several heritage buildings and shrines, has a very high level of
structural vulnerability and as we move towards the outer peripheral wards the vulnerability
changes from moderate to low. The probability of masonry buildings collapsing in the event
of an earthquake is higher (Bhosale et al.,  2018), and the city has a large percentage of such
buildings, making it more vulnerable to earthquake disasters. Buildings with regular
geometry, uniform mass distribution and rigidity in plan and elevation are more resistant to
earthquakes than buildings with irregular geometry and hence variable mass distribution
(Stein, 1982). As the findings of this study show, a good number of buildings in a few wards
of the city have irregular geometry, making them more vulnerable to earthquakes. The high
building density, maximal pounding potential and narrower road network near the city centre
part make these wards particularly vulnerable when compared to the other wards located in
the periphery of the city.

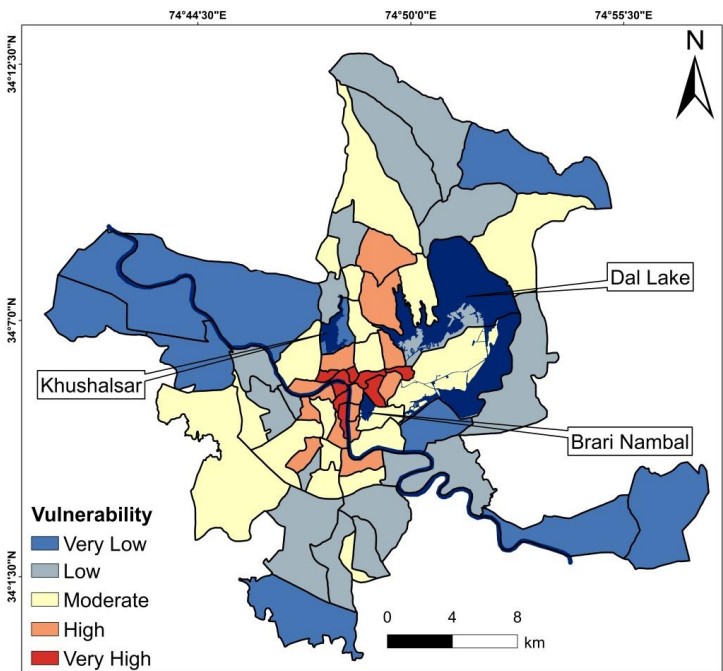

Fig. 10. Structural vulnerability of Srinagar city.

Since Because the majority of the built-up in the city is non-engineered, highly dense, irregular and masonry based, the results indicate that infrastructure development of any type in the very high and high vulnerable zones of the city must adhere to the prescribed building codes and bylaws to achieve the resilience to earthquakes. However, the continued construction of both government and residential buildings in the wetlands and marshy areas of Srinagar city, particularly towards the south of the city, is worrisome because it makes the city more vulnerable to the earthquake damage. Furthermore, in the event of a big earthquake, the lack of key amenities such as trauma hospitals, shelters etc. and poor road conditions in several wards of Srinagar city might cause significant damage to life and property. Earthquake vulnerability assessment of the built-up environment in Srinagar, if followed by retrofitting, restoration and rehabilitation initiatives in the most vulnerable wards of the city will help to reduce damage during earthquakes. In very high and high vulnerable zones, provision for emergency services such as firefighters, shelters, specialized medical facilities and so on must be made to minimize the loss of life and property in the event of an earthquake. Pre- and post-earthquake disaster mitigation and capacity-building initiatives are critical for transforming Srinagar into a safe, sustainable and earthquake-resistant city. The




challenges surrounding the earthquake threat to Srinagar and the city's preparedness thereof necessitates the adoption of new scientific and innovative urban development planning and inexpensive measure aimed at inculcating a culture of earthquake consciousness among its citizenry. The establishment of a culture of earthquake-resistant and safe constructions will undoubtedly make the city safer and reduce the adverse consequences of earthquakes.

## 5.  Conclusions

Understanding the structural vulnerability of a city situated in an earthquake-prone zone  at a ward scale is critical for deciding on appropriate urban planning and development strategies to build and promote a safe, inclusive, sustainable, and earthquake-resilient living environment as contempled under SDG 11.  The current study, which is the first of its kind for Srinagar, reveals the micro-level structural vulnerability of the built-up environment in the city. The vulnerability zonation map generated for the city reveals that around 32% of the city has a very low vulnerability, which covers 10 municipality wards. The low earthquake vulnerability zone encompasses around 33% of the city, which includes 17 wards; the moderate vulnerability zone covers around 28% of the city, and 19 wards; the high vulnerability zone covers 5.7 % of the city, and 14 wards and the very high vulnerability zone covers 1.28 % of the city and 9 municipality wards. Overall, about 7% of the city covering $1/3^{rd}$ of the city municipal wards (n=23) are falling into either high or very high vulnerability zones. The downtown wards in the city's central area are the most vulnerable to earthquakes due to the high pounding potential, high building density, and smaller streets with little or no open and green areas. Since green and open spaces are used as evacuation places, therefore, it is strongly advised that new constructions in these areas as well as the development of these spaces, must be avoided. The study underlines the importance of developing emergency action plans that outline how to prevent causalities by allowing for the rapid, selective and effective utilisation of resources as well as retrofitting schemes and capacity-building programs to safeguard human life and the economy in the city. These findings are consistent with the posteriori knowledge of the study area's vulnerability and they will help the urban planners and policymakers in developing any future land use planning and strategies. The socio-economic vulnerability of the city was not analysed in this study, but it would be included in future research to produce a more accurate and holistic assessment of the earthquake vulnerability to better inform policymaking for developing earthquake risk reduction strategies in the city.





**Author contributions:**

**Shakil Ahmad Romshoo:** Conceptualization, Methodology, Supervision. Manuscript preparation with inputs from **MF, Midhat Fayaz:** Data generation, Methodology, Formal analysis, Field surveys, Investigation, **Irfan Rashid:** Investigation, Review and Editing, **Rakesh Chandra**: Investigation, Review and Editing.

**Competing interests**: "The authors declare that they have no conflict of interest."

**Funding:**

The work was funded by Ministry of Earth Sciences (MoES), Govt. of India, New Delhi, under the award number MoES/P.O. (Geosci)/16/ 2013. The financial assistance received from the sponsors under the project is thankfully acknowledged.

**Acknowledgement:**

The research work was conducted under the Ministry of Earth Sciences sponsored research project titled "Geological characterization of the Kashmir valley with the objective of quantifying probabilistic hazard and risk in high risk areas of the valley using a logically integrated set of geoscientific investigations". The financial assistance received from the sponsors under the project is thankfully acknowledged.

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
