# Peer review of "Earthquake Vulnerability Assessment of the Built Environment in Srinagar City,"

_Natural Hazards and Earth System Sciences, 2022_

## Author Comment (AC1)

**COMMUNITY COMMENT #1**

**General comment:** This manuscript is very interesting and on an important topic of earthquake vulnerability assessment in one of the most vulnerable regions of the Indian Himalayas. The topic has high significance and the approaches used are good. Though well written there are some typos in the paper that need to be corrected. However, it may help the authors to improve the manuscript by considering my following comments and suggestions:

**Response:** Thank you for the very useful comments and suggestions, which have helped us to improve the contents of the manuscript. We have responded point by point to all the comments and suggestions raised by the Commentator#1 as follows:

**Comment #1:** The authors have provided useful information about the past destructive earthquakes in the region in a tabular form. Despite such a long history of destructive earthquakes, the authors claim that people don't follow the building codes and plans which is hard to believe.

**Response:** Thanks for the appreciation. Yes, it is a fact that despite the building codes being in place, but the same are not followed in letter and spirit by the people due to the poor implementation of the regulations and inadequate monitoring mechanisms (Yousuf et al., 2020). However, this is a scenario in most cities of the developing countries (Petal et al., 2008). The residential buildings in the Srinagar city are mostly built by local semi-skilled masons, who don't have the adequate technical expertise in building earthquake resistance infrastructure and therefore these structures lack the basic earthquake risk reduction features including seismic resistance features as are otherwise prescribed in the building codes. Furthermore, economy plays a major role in constructing safe infrastructure and there are socio-economic inequalities within the city, with the citizens in the lower-middle-income groups residing in the densely populated old downtown wards which have grown organically without any urban planning or adherence to building codes. Whereas the upper-middle class and wealthy people reside in the uptown wards of the city where, building codes and other infrastructure development regulations are usually enforced as per the master plan prescriptions. The same has been briefly mentioned in the revised manuscript from line number 74 to 78.

Yousuf, M., Bukhari, S. K., Bhat, G. R., and Ali, A.: Understanding and managing earthquake hazard visa viz disaster mitigation strategies in Kashmir valley, NW

Himalaya, Progress in Disaster Science, 5, 100064, https://doi.org/10.1016/j.pdisas.2020.100064, 2020.

Petal, M., Green, R., Kelman, I., and Shaw, R. (2008). Community-based construction for disaster risk reduction. In *Hazards and the built environment* (pp. 209-235). Routledge.

Comment#2: The authors should provide information about urban/city planning of the region and why these are ineffective (in the Study area section) and State briefly how findings from this research might help to build better earthquake-safe master plan for the city (in the Conclusion section).

**Response:** Thank you for your suggestions. Srinagar is an old and historic city, and most of the areas have grown organically without following any physical plan or building codes for the construction of its built infrastructure (Yousuf et al., 2020). Post-1947, Srinagar grew very fast mostly in a haphazard manner with no proper urban planning. The first Master Plan of the city was developed in 1971 followed by Master Plan-2021 and 2035. However, all the three plans didn't have effective implementation in the city as per the Master Plan prescriptions because of the problems in the planning and implementation setup including the inadequate legal framework and institutional structures. The chronology of the city master planning and why it has not been very effective is provided in the revised manuscript under study area section from line number 163 to 167.

Definitely, this study can assist city planners in choosing safe, low-density areas, and even guide to propose new infrastructural development envisaged under the master plan, as well as identify densely populated areas that are particularly vulnerable to earthquakes where no further infrastructural development should be permitted other than the development of open and green spaces. The same has been incorporated in the revised manuscript from line number 584 to 588.

**Comment #3:** The authors have used two methods; AHP for assigning weights and MCA-based TOPSIS for ranking wards based on the best alternatives. The integration of the outcome from these two methods is not very clear and needs to be elaborated further in the methods section.

**Response:** Thank you for suggestion, we have added information about the advantage of the integrative use of the two approaches in the revised manuscript from line number 390 to 399 and the same is reproduced as follows:

The integrative use of these two models reduces the uncertainty in the input data and improves accuracy and validity. Furthermore, decision-making based on the integrated use of the AHP and TOPSIS leads to more robust and effective outcomes for addressing complex problems (Nyimbili et al., 2018). Many studies have recommended the integrated use of TOPSIS with AHP for determining criteria and conducting analyses regarding complex decision-making problems (Behzadian et al., 2012). Additionally, the integrated use of AHP and TOPSIS helps to resolve the weighting problem by incorporating expert opinions and preferences, thereby increasing the consistency of outputs for arriving at consensus in decision-making in earthquake disaster vulnerability analyses (Nyimbili et al., 2018).

Nyimbili, P. H., Erden, T., and Karaman, H.: Integration of GIS, AHP and TOPSIS for earthquake hazard analysis, Natural hazards, 92(3), 1523-1546. https://doi.org/10.1007/s11069-018-3262-7, (2018)

Behzadian, M., Otaghsara, S. K., Yazdani, M., Ignatius, J.: A state-of the-art survey of TOPSIS applications, Expert Systems with Application, 39(17):13051–13069 https://doi.org/10.1016/j.eswa.2012.05.056, 2012.

**Comment #4:** The authors have, at a few places in the paper, briefly talked about the virtues of the traditional wooden earthquake-resistant construction practices in the city but have not really provided any details about these traditional buildings and how these are considered earthquake resistant. It would add value to the manuscript if a section is added in the paper on these traditional and now abandoned construction types

**Response:** Thank you for the suggestion. We have now included the details of the traditional construction practices and why these are considered resistant to earthquakes in the revised manuscript under introduction section between the lines number 65 to 72 and the same is reproduced as follows:

The traditional building types such as "Taqq" and "Dhajji-Dewari" are earthquake-resistant. In the Taqq type buildings, wooden runners are placed at each floor level that tie the walls with the floor together whereas the Dhajji-Dewari buidlings consist of a braced timber frame with masonry infill that is placed diagonally in the walls. The timber braced frames offer stable confinement to the infill masonry as long as it rests together (Hicyilmaz et al., 2012). When compared to more contemporary building types, the Dhajji-Dewari constructions are more earthquake-resistant because energy is dissipated between mortar joints, the frame, and the infill rather than through non-linear deformations.

Hicyilmaz, K. M. O., Wilcock, T., Izatt, C., Da Silva, J., and Langenbach, R.: Seismic performance of dhajji dewari. In 15th World Conference on Earthquake Engineering, Lisbon (pp. 24-28). 2012.

**Comment #5:** The authors have not included socio-economic vulnerability in this paper and intend to do that as separate research; however, it would be helpful to the readers if the authors can provide a brief general overview of the SE vulnerability in this paper.

**Response:** Thank you for the suggestion. We have now added a general overview of the SE vulnerability of the Srinagar city in the revised manuscript under section Earthquake vulnerability analysis at lines 566 to 578.

The socio-economic conditions of an area play an imporant role in determining the vulnerability of an area to earthquake hazard. The Srinagar city has witnessed population explosion with the population having increased from 0.25 million in 1961 to 1.5 million in 2011. The city also has a high percentage of female and child population (59%) and a high population density of 4000 per sq.km. Migration from rural areas and population growth are the primary drivers of this enhanced population expansion (Nengroo et al., 2018). The city has been under pressure to expand its built-up area in order to cater to the population boom, which has also led to excessive resource depletion, widening wealth and poverty gaps, and detrimental environmental and socioeconomic concerns (Mitsovaa et al., 2010; Kamat and Mahasur, 1997). With the mounting demand for new housing, the quality and condition of houses have received negligible attention. These concerns of accelerated population progression, along with high urbanisation have increased the Socio-economic vulnerability of the built environment in the Srinagar city to earthquakes.

Nengroo, Z. A., Bhat, M. S., and Kuchay, N. A.: Measuring urban sprawl of Srinagar city, Jammu and Kashmir, India, Journal of Urban Management, 6(2), 45-55. https://doi.org/10.1016/j.jum.2017.08.001, 2017.

Mitsova, D., Shuster, W., and Wang, X.: A cellular automata model of land cover change to integrate urban growth with open space conservation, Landscape and urban planning, 99(2), 141-153. https://doi.org/10.1016/j.landurbplan.2010.10.001 2011.

Kamat, S. R., and Mahasur, A. A.: Air pollution: slow poisoning Chennai, The Hindu Survey of Environment, 1997.

---

## Author Comment (AC2)

**COMMUNITY COMMENT #2**

**General comment:** The manuscript is well written and valuable work. The authors have given a good overview of the earthquake history of Kashmir region. Analysis is also good and conclusions are also meaningful. I found this paper interesting and therefore I am making few suggestions which are given below and may be considered by the authors.

**Response:** Thank you for your appreciation, insightful comments and suggestions. The suggestions helped us to improve contents and structure of the manuscript significantly.

**Comment 1:** -You have mentioned SDG-11 in the introduction and conclusion. But elaborate about it more in the conclusion as to how this study will help in achieving the SDG-11

**Response:** Thank you for the suggestion. We have added more details about the SDG-11 in the revised manuscript from line number 621 to 628 under Conclusion section.

The current study is in accordance with the 2030 Agenda for Sustainable Development Goals, which recognises and reiterates the urgent need to lower the risk of disasters. The study will help to reduce the exposure and vulnerability of people to disasters and build resilient infrastructure. The findings of this study will support sensible urban planning, which calls for the construction of resilient infrastructure to reduce vulnerability to natural disasters, as well as sustainable development in line with the SDG 11 and SDG 9, which demand manageable densities, user-friendly public spaces, and mixed-use urban development.

**Comment 2:** - I would suggest adding limitation of the two models in the paper. Other than the authors, who were the people involved in the expert judgement process.

**Response:** Thank you for the suggestion. We have added it in the revised manuscript line number 400 to 405.

The adopted methodology has a few limitations, much like any other modelling technique. In addition to the inherent flaws in Multi Criteria Decision Analysis (MCDA), there may be some limitations, such as the fact that certain layers become more dominant than others due to the weighting criteria used, which in turn depends upon the decision-makers' perceptions of which vulnerability parameters have the greatest impact on modelling outcomes in vulnerability analysis.

**Comment 3:** - Make a mention of these people and their expertise.

**Response:** Thank you for the suggestion. Though, only the four authors were involved in determining the expert judgement process, viz., Prof. Shakil Ahmad Romshoo, Ph.D Remote Sensing and GIS, Dr. Irfan Rashid Ph.D Environmental Sciences, Dr. Rakesh Chandra, Ph.D Geology and Midhat Fayaz, M.Sc. (Geoinformatics) but a large body of literature was also consulted that informed the expert judgement process. The same has been mentioned in the revised manuscript, line number 336 to 340.

**Comment 4:** References of some methods are missing, for example separation distance, closeness etc.

**Response:** Thanks for the comment. We have added the references for these methods in the revised manuscript.

**Comment 5:** - The building density in some areas of the city is shown very sparse (Fig. 8). I do not know if these are high-altitude hilly area or forests and other uninhabitable area or whatever. The authors should drape the building density map on a DEM or other elevation/topographic data/forest area/whatever of the wards/zones be provided

**Response:** Thanks for this suggestion. The elevation profile of the city is already shown in fig. 2 (a). Srinagar is one of the largest urban centre in the Himalayan region and is experiencing considerably high rates of population growth and built-up area expansion, leading to the extension of urban areas and the merging of the city fringe areas into the main city (Bhat et al., 2012). The outer peripheral wards have mostly low building density as these are the developing areas proposed under Srinagar Master Plan 2035. The same has been mentioned in the revised manuscript at line number 481 to 486.

Bhat, S., Ahmad, A., Bhat, M. S., Zahoor, A. N., Kuchay N. A., Bhat, M. S., Mayer A. I., and Sabar, M.: Analysis and simulation of urban expansion of Srinagar city, *Sciences*, *2249*, 2224-5766. https:// doi.org/ 10.5897/IJPC2015.0314, 2012.

**Comment 6:** -Why some wards, which have almost very low building density, are having moderate vulnerability. Compare Fig. 8 and Fig. 10.

**Response:** Thank you very much for this comment. This is because each vulnerability criteria varies in importance, magnitude and impact, despite these wards' low building density; they also have the highest proportion of masonry and irregularly shaped buildings, with a

moderately dense road network, which increases their susceptibility to vulnerability. The figure below shows the weightage of different indicators of built-up environment vulnerability

[Figure]

Fig 1: Influence of parameters on earthquake vulnerability of Srinagar city.

**Comment 7:**-I see that the authors have masked the water bodies in the city from analysis. Does it include marshy lands and wetlands also. I see that you have not used wetland/marshes in the analysis, if, I am correct.

**Response:** - Thanks for the comment. Yes, we have not used the wetlands/marshy lands in the analysis and the same is mentioned in the revised manuscript at line number 564 to 565.

**Comment 8:**- There are several typos in the paper which need to be corrected.

**Response:** - Thanks for pointing out this and we have corrected all the typos and grammar in the revised manuscript.

---

## Author Comment (AC3)

**REFREE COMMENTS # 1**

**General comment:** This manuscript is about earthquake vulnerability assessment in the Indian Himalayas. The topic and techniques are good. The paper is well-written yet has errors. Consider my remarks and ideas to improve the manuscript.

**Response:** Thank you very much for appreciating the work. We felt gratitude to you for your thoughtful review of our work. We have now carefully revised the manuscript in light of your comments and suggestions and the point-by-point response to your comments and suggestion is provided below:

**Comment 1:**-The authors utilized AHP for weighting and MCA-based TOPSIS for ranking wards. The integration of these two strategies has to be clarified in the methods section.

**Response:** Thank you for suggestion, we have added information about the advantage of the integrative use of the two approaches in the revised manuscript from line number 390 to 399 and the same is reproduced below:

The integrative use of these two models reduces the uncertainty in the input data and improves accuracy and validity. Furthermore, decision-making based on the integrated use of the AHP and TOPSIS leads to more robust and effective outcomes for addressing complex problems (Nyimbili et al., 2018). Many studies have recommended the integrated use of TOPSIS with AHP for determining criteria and conducting analyses regarding complex decision-making problems (Behzadian et al., 2012). Additionally, the integrated use of AHP and TOPSIS helps to resolve the weighting problem by incorporating expert opinions and preferences, thereby increasing the consistency of outputs for arriving at consensus in decision-making in earthquake disaster vulnerability analyses (Nyimbili et al., 2018).

Nyimbili, P. H., Erden, T., and Karaman, H.: Integration of GIS, AHP and TOPSIS for earthquake hazard analysis, Natural hazards, 92(3), 1523-1546. https://doi.org/10.1007/s11069-018-3262-7, 2018

Behzadian, M., Otaghsara, S. K., Yazdani, M., Ignatius, J.: A state-of the-art survey of TOPSIS applications, Expert Systems with Application, 39(17):13051–13069 https://doi.org/10.1016/j.eswa.2012.05.056, 2012.

**Comment 2:**-Nevertheless, it would be helpful to the readers if the authors could include a quick general review of the SE vulnerability in this study. The authors did not include socio-economic vulnerability in this paper since they want to do that as part of another research.

**Response:** Thank you for the suggestion. As suggested, we have now added a general overview of the SE vulnerability of the Srinagar city in the revised manuscript under the section "Earthquake vulnerability analysis" from lines 566 to 578 and the same is reproduced below for your perusal:

The socio-economic conditions of an area play an imporant role in determining the vulnerability of an area to earthquake hazard. The Srinagar city has witnessed population explosion with the population having increased from 0.25 million in 1961 to 1.5 million in 2011. The city also has a high percentage of female and child population (59%) and a high population density of 4000 per sq.km. Migration from rural areas and population growth are the primary drivers of this enhanced population expansion (Nengroo et al., 2018). The city has been under pressure to expand its built-up area in order to cater to the population boom, which has also led to excessive resource depletion, widening wealth and poverty gaps, and detrimental environmental and socioeconomic concerns (Mitsovaa et al., 2010; Kamat and Mahasur, 1997). With the mounting demand for new housing, the quality and condition of houses have received negligible attention. These concerns of accelerated population progression, along with high urbanisation have increased the Socio-economic vulnerability of the built environment in the Srinagar city to earthquakes.

Nengroo, Z. A., Bhat, M. S., and Kuchay, N. A.: Measuring urban sprawl of Srinagar city, Jammu and Kashmir, India, Journal of Urban Management, 6(2), 45-55. https://doi.org/10.1016/j.jum.2017.08.001, 2017.

Mitsova, D., Shuster, W., and Wang, X.: A cellular automata model of land cover change to integrate urban growth with open space conservation, Landscape and urban planning, 99(2), 141-153. https://doi.org/10.1016/j.landurbplan.2010.10.001 2011.

Kamat, S. R., and Mahasur, A. A.: Air pollution: slow poisoning Chennai, The Hindu Survey of Environment, 1997.

**Comment 3:**-In the paper, I think it would be helpful to include a restriction of each of the two models.

**Response:** Thank you for the suggestion. We have added limitations of the two models in the revised manuscript line number 400 to 405 which is reproduced below:

The adopted methodology has a few limitations, much like any other modelling technique. In addition to the inherent flaws in Multi Criteria Decision Analysis (MCDA), there may be some limitations, such as the fact that certain layers become more dominant than others due to the weighting criteria used, which in turn depends upon the decision-makers' perceptions of which vulnerability parameters have the greatest impact on modelling outcomes in vulnerability analysis.

**Comment 4:**- Who, besides the writers, took part in the process of making the expert judgement, and what were their qualifications? Mention these individuals and the expertise that they bring to the table.

**Response:** Thank you for the comment. Though, only the four authors were involved in determining the expert judgement process, viz., Prof. Shakil Ahmad Romshoo, Ph.D Remote Sensing and GIS, Dr. Irfan Rashid Ph.D Environmental Sciences, Dr. Rakesh Chandra, Ph.D Geology and Midhat Fayaz, M.Sc. (Geoinformatics) but a large body of literature was also consulted that informed the expert judgement process. A mention of the same has been made in the revised manuscript at line number 336 to 340.

**Comment 5:**- There is a lack of references to certain procedures, such as those pertaining to proximity, closeness, and separation.

**Response:** Thanks for the comment. We have added references for all the approaches used in paper in the revised manuscript.

**Comment 6:**-Both in the introduction and in the end, you made a reference to SDG-11. In the conclusion, however, you should go into more detail on how the findings of this study will contribute to the achievement of SDG-11.

**Response:** Thank you for the suggestion. We have added more details about the SDG-11 in the revised manuscript from line number 621 to 628 under Conclusion section.

The current study is in accordance with the 2030 Agenda for Sustainable Development Goals, which recognises and reiterates the urgent need to lower the risk of disasters. The study will help to reduce the exposure and vulnerability of people to disasters and build resilient infrastructure. The findings of this study will support sensible urban planning, which calls for the construction of resilient infrastructure to reduce vulnerability to natural disasters, as well as sustainable development in line with the SDG 11 and SDG 9, which demand manageable densities, user-friendly public spaces, and mixed-use urban development.

**Comment 7:**- There are several typos in the paper which need to be corrected.

**Response:** Thank you very much for this suggestion. We have now rectified all the typos and grammar errors in the revised manuscript

---

## Author Comment (AC4)

**REFREE COMMENTS # 2**

**General comment:** Midhat Fayaz et al. present a vulnerability assessment of the buildings in Srinagar city in the event of earthquake. The authors consider 69 municipal wards within Srinagar urban area. The analysis is based on a building inventory having details of the nature and structure of buildings. The authors perform a ground-based survey to validate the inventory which is commendable. The manuscript offers some insights into differential vulnerabilities of buildings within the urban area. Though there is not any visible flaw in the analysis and assessment some fundamental aspects of earthquake science is missing in the manuscript. Therefore, major revisions as following are required for this manuscript to be considered for publication.

**Response:** Many thanks for commending our work. Authors express their gratitude to the Reviewer for the careful assessment of our work and for valuable suggestions and comments, the incorporation of which have improved the quality of the revised manuscript. We agree with the comment that some of the earthquake related parameters are not included in this study. Below is the point-by-point response to the comments/suggestions.

**Comment 1:**-The authors list major earthquake events in Srinagar city/Kashmir. However, they do not explain how the relate the earthquake related parameters i.e., epicenter with the vulnerability assessments.

**Response:** Thank you for the comment. Agreed that the location of earthquake epicentre indicates the presence of geological structures (faults) in a particular area (Sana, 2018). The available records of historical and instrumental earthquake events (Table 1) in the study area indicate a high probability of earthquake events in the Srinagar city in the future. Dar et al., 2019 have shown that the River Jhelum, running through Srinagar city itself flows along or parallel at many places to a lineament or fault known as Jhelum fault in the Kashmir Valley. Besides high tectonic activity and the lithology (mostly unconsolidated sediments) of the area, makes the area vulnerable to earthquakes.

Therefore, it is believed that in light of the high vulnerability and occurrences of past earthquakes with epicentre in and around Srinagar makes the earthquake vulnerability assessment of study area an important exercise irrespective of the exact location of the epicentre. We assumed that the entire city is vulnerable because of the presence of the Jhelum fault in the midst of the city.

We have modified the study area map in the revised manuscript showing the nearest faults/lineament details (see fig. 1) and made a mention of the above facts in the revised manuscript at line number 124-138.

Sana, H.: Seismic microzonation of Srinagar city, Jammu and Kashmir, Soil Dynamics and Earthquake Engineering, 115, 578-588, https://doi.org/10.1016/j.soildyn.2018.09.028, 2018.

Dar, R. A., Mir, S. A., and Romshoo, S. A.: Influence of geomorphic and anthropogenic activities on channel morphology of River Jhelum in Kashmir Valley, NW Himalayas, Quaternary International, 507, 333-341, https://doi.org/10.1016/j.quaint.2018.12.014, 2019.

[Figure]

Fig.1: Location of the study area. Here MBT stands for Main Boundary Thrust, MCT stands for Main Central Thrust, BF stands for Balapur Fault.

**Comment 2:**- Nowhere it is mentioned how does the geology, lithology and faults are considered to zonate the likelihood of earthquake events within the city. It would have been helpful if the authors have at least considered an earthquake hazard zonation map in their analysis. Without these it is surprising that how the authors assess the earthquake vulnerability of buildings. The present manuscript gives an impression of vulnerability of buildings alone but not for earthquake events.

**Response:** Thanks for the comment. The present study was carried out to look into the vulnerability of built-up environment at a high spatial resolution at building footprint scale ward-wise in the Srinagar city. Geology, lithology, and lineaments faults were taken into account during data analysis. However, keeping in view the high occurrences of the past earthquakes with epicentre in and around Srinagar irrespective of the exact location of the epicentre and distribution of other geological/geomorphic and soil parameters, the entire city wards are equally vulnerable to earthquakes in the eventuality of an earthquake. The River Jhelum, running through the Srinagar city, itself flows along or parallel at many places to a lineament or fault known as Jhelum fault in the Kashmir Valley. Besides because of the high tectonic activity and the lithology (mostly unconsolidated sediments) of the area, it was found that there is very little difference in earthquake vulnerability between various wards because of the similar tectonic, lithologic and geomorphic set up of the city wards; therefore all of these parameters were kept constant. All of Srinagar's wards are situated on consolidated alluvium, or Karewas, which share similar characteristics in terms of how they react to earthquakes. Panjal volcanics are located in a few inconspicuous places, however these are hills that have no habitation, making lithology the least influential parameter in this study. Additionally, there is a very high earthquake risk in each of Srinagar's wards (Sana, 2018; Yousuf and Bukhari, 2020) (see Fig.2). Since the vulnerability at the ward level is the primary focus of the current study, all of the wards were treated as having an equal risk from seismic activity. Please see figure 2 and Figure 3 below for details about the distribution of earthquake related parameter and earthquake hazard map.

Accordingly, we have made a mention of same in the revised manuscript from line number 124-138 under Introduction Section.

Sana, H.: Seismic microzonation of Srinagar city, Jammu and Kashmir, Soil Dynamics and Earthquake Engineering, 115, 578-588, https://doi.org/10.1016/j.soildyn.2018.09.028, 2018.

Yousuf, M., Bukhari, S. K., Bhat, G. R., and Ali, A.: Understanding and managing earthquake hazard visa viz disaster mitigation strategies in Kashmir valley, NW Himalaya, Progress in Disaster Science, 5, 100064, https://doi.org/10.1016/j.pdisas.2020.100064, 2020.

[Figure]

Fig.2: lithology map modified after Thakur and Rawat, 1992.

[Figure]

Fig.3: Seismic Hazard Index map modified after Sana, 2018.

**Comment 3:-** In addition, buildings are susceptible to failures due to ground failure during earthquakes i.e., liquefaction and other soil-structure related damages. Nowhere in this study these aspects have been mentioned or considered.

**Response:** Thanks for the comment. In a similar fashion like that of the lithology, soils and lineaments, the Liquefaction Potential Index (LPI) of the Srinagar shows very least variability from one ward to another. Please see the Figure 4 below for the Liquefaction Potential Index

of the Srinagar city (Sana et al., 2016). All the wards of the city fall within LPI of high to very high zone. Thus, like other geological factors, this factor was also kept constant for all the wards of Srinagar under study. It is pertinent to mention here that there are several studies conducted globally that have not included the geological/earthquake parameters in the analysis of earthquake vulnerability assessment of built up environment of cities because of the similar reasons e.g. Srikanth et al., 2010; Ishita and Khandaker 2010; Islam et al., 2013; Alizadeh et al., 2018; Adhikari et al., 2019; Menegon et al., 2019; Fan et al., 2021.

In light of the Reviewer's comments and above explanations, we have made a mention of the above response in the revised manuscript at line number 124-138.

[Figure]

Fig.4: Liquefaction Potential Index map modified after Sana, 2016.

Sana, H., and Nath, S. K. (2016). Liquefaction potential analysis of the Kashmir valley alluvium, NW Himalaya. *Soil Dynamics and Earthquake Engineering*, *85*, 11-18.https://doi.org/10.1016/j.soildyn.2016.03.009

Srikanth, T., Kumar, R. P., Singh, A. P., Rastogi, B. K., and Kumar, S. (2010). Earthquake vulnerability assessment of existing buildings in Gandhidham and Adipur cities Kachchh, Gujarat (India). European Journal of Scientific Research, 41(3), 336-353.

Ishita, R. P., and Khandaker, S. (2010). Application of analytical hierarchical process and GIS in earthquake vulnerability assessment: case study of Ward 37 and 69 in Dhaka City. J Bangladesh Inst Plan ISSN, 2075, 9363.

Islam, M. S., Sultana, N., Bushra, N., Banna, L. N., Tusher, T. R., and Ansary, M. A. (2013). Effects of earthquake on urbanization in Dhaka City. Journal of Environmental Science and Natural Resources, 6(1), 107-112.

Alizadeh, M., Hashim, M., Alizadeh, E., Shahabi, H., Karami, M. R., Beiranvand Pour, A., ... and Zabihi, H. (2018). Multi-criteria decision making (MCDM) model for seismic vulnerability assessment (SVA) of urban residential buildings. ISPRS International Journal of Geo-Information, 7(11), 444.

Adhikari, A., Rao, K. R. M., Gautam, D., and Chaulagain, H. (2019). Seismic vulnerability and retrofitting scheme for low-to-medium rise reinforced concrete buildings in Nepal. Journal of Building Engineering, 21, 186-199.

Menegon, S. J., Tsang, H. H., Lumantarna, E., Lam, N. T. K., Wilson, J. L., and Gad, E. F. (2019). Framework for seismic vulnerability assessment of reinforced concrete buildings in Australia. Australian Journal of Structural Engineering, 20(2), 143-158.

Fan, X., Nie, G., Xia, C., and Zhou, J. (2021). Estimation of pixel-level seismic vulnerability of the building environment based on mid-resolution optical remote sensing images. International Journal of Applied Earth Observation and Geoinformation, 101, 102339.

---

## Author Response (AR1)

**Editor Report**

**General Comment:** Two experts have evaluated the manuscript "nhess-2022-155': Earthquake Vulnerability Assessment of the Built Environment in Srinagar City, Kashmir Himalaya, Using GIS. The author(s) have presented an earthquake vulnerability assessment of the buildings in Srinagar city. The analysis is based on a building inventory having details of the nature and structure of buildings. The authors perform a ground-based survey to validate the inventory which is commendable.

**Response:** We express our gratitude to the Editor, the two anonymous reviewers and the two community commentators for the detailed and useful review of our manuscript. The comments and suggestions by the Editors, reviewers and commentators have significantly improved the quality of the revised manuscript. We have addressed all the concerns of the reviewers, and incorporated all the suggestion in the revised manuscript (marked in track change mode in the revised manuscript). As a result, the revised manuscript is much improved both in content and structure

**Comment**: Both referees agreed that the manuscript addresses a relevant topic that might be useful to the readers of NHESS and relevant stakeholders. However, both referees had made very important and fundamental critics, mainly doubts concerning the adequacy of the proposed method (integration of AHP for weighting and MCA-based TOPSIS for ranking wards; no explanation on how to relate the earthquake-related parameters i.e., epicenter with the vulnerability assessments; provide earthquake hazard zonation map), the soundness of the conclusions and novelty (see review 1 and 2 comments).

**Response**: Thank you very much for highlighting the relevance of our work for the journal and its importance to the relevant stakeholders. All the concerns of the referees have been addressed and responded in detail in the Author's Response to the individual Reviewers and Commentators. The clarification regarding the methodology of integrating AHP for weighting and MCA-based TOPSIS for ranking wards was raised by one of the Reviewers as well as a Commentator and the same was responded to and a clarification has been now included in the revised manuscript from line no. 390 to 399.

The query regarding the relation between epicentre and vulnerability assessment by another Reviewer has been responded in the Author's response to the Reviewer and is addressed in the revised manuscript from line no. 124 to 138. We have also modified the figure 1 of the

manuscript in light of the comment from the same Reviewer and the figure now shows the location of geological structures like major faults located in and around the study area. In response to the another comment of Referee II, we have produced the seismic hazard map of the Srinagar city along with lithology map and liquefaction potential map in Author's response. It is pertinent to mention here that the seismic hazard index of the Srinagar city is very high to severe and shows very little ward to ward variation. We have modified the conclusion section to increase its soundness and also described the novelty of the work in the revised manuscript from line no. 614 to 616 and 621 to 628.

**Comment**: Both referees had asked for additional explanations, justifications, and analysis concerning: a review of the SE vulnerability; limitation of two models; qualifications and details of expert judgment; fundamental aspects of the earthquake and others.

**Response**: Thanks for the reiteration of the Reviewers' and Commentators' comments. A general overview of the SE of the Srinagar city has been included in the revised manuscript from line no. 566 to 576. We have highlighted the general limitations of the two models AHP and TOPSIS in the revised manuscript from line no. 400 to 405. The line no. 336 to 340 of the revised manuscript describes the qualifications and details of the experts involved in expert judgement process.

**Comment**: Both the referees have strongly indicated plenty of grammatical mistakes and poor readability.

**Response**: Thank you very much for again highlighting the need for correcting the typos and grammatical errors in the manuscript. We have now corrected these mistakes and typos which has significantly improved the readability of the manuscript.

**Comment**: The Authors gave responses to some of the main Referees' comments. Both referees recommend major revisions of the paper. The paper will be reconsidered after "Major revisions". In the revised version, authors are invited to respond point by point to each referee's comment, giving convincing proof to all questions raised by referees.

**Response**: We appreciate the decision of the Editor for a major revision of the manuscript. Accordingly, we have responded point-by-point in our Author's Response to all the comments and suggestions of the two Reviewers and two Commentators providing all the

explanations, required proofs and important references. Thereafter, the suggestions were incorporated in the respective appropriate sections of the revised manuscript which has improved the quality of the revised manuscript in terms of content and structure.

**Comment**: Please highlight clearly what you changed in the revised manuscript so the referees are able to assess your changes.

Response: Thanks, we have clearly highlighted all the changes made in the revised manuscript, in response to the comments and suggestions of the Reviewers' and Commentators' to make these modifications easily trackable for the editor and the reviewers.

**REFREE COMMENTS # 1**

**General comment:** This manuscript is about earthquake vulnerability assessment in the Indian Himalayas. The topic and techniques are good. The paper is well-written yet has errors. Consider my remarks and ideas to improve the manuscript.

**Response:** Thank you very much for appreciating the work. We felt gratitude to you for your thoughtful review of our work. We have now carefully revised the manuscript in light of your comments and suggestions and the point-by-point response to your comments and suggestion is provided below:

**Comment 1:-**The authors utilized AHP for weighting and MCA-based TOPSIS for ranking wards. The integration of these two strategies has to be clarified in the methods section.

**Response:** Thank you for suggestion, we have added information about the advantage of the integrative use of the two approaches in the revised manuscript from line number 390 to 399 and the same is reproduced below:

The integrative use of these two models reduces the uncertainty in the input data and improves accuracy and validity. Furthermore, decision-making based on the integrated use of the AHP and TOPSIS leads to more robust and effective outcomes for addressing complex problems (Nyimbili et al., 2018). Many studies have recommended the integrated use of TOPSIS with AHP for determining criteria and conducting analyses regarding complex decision-making problems (Behzadian et al., 2012). Additionally, the integrated use of AHP and TOPSIS helps to resolve the weighting problem by incorporating expert opinions and preferences, thereby increasing the consistency of outputs for arriving at consensus in decision-making in earthquake disaster vulnerability analyses (Nyimbili et al., 2018).

Nyimbili, P. H., Erden, T., and Karaman, H.: Integration of GIS, AHP and TOPSIS for earthquake hazard analysis, Natural hazards, 92(3), 1523-1546. https://doi.org/10.1007/s11069-018-3262-7, 2018

Behzadian, M., Otaghsara, S. K., Yazdani, M., Ignatius, J.: A state-of the-art survey of TOPSIS applications, Expert Systems with Application, 39(17):13051–13069 https://doi.org/10.1016/j.eswa.2012.05.056, 2012.

**Comment 2:**-Nevertheless, it would be helpful to the readers if the authors could include a quick general review of the SE vulnerability in this study. The authors did not include socio-economic vulnerability in this paper since they want to do that as part of another research.

**Response:** Thank you for the suggestion. As suggested, we have now added a general overview of the SE vulnerability of the Srinagar city in the revised manuscript under the section "Earthquake vulnerability analysis" from lines 564 to 576 and the same is reproduced below for your perusal:

The socio-economic conditions of an area play an important role in determining the vulnerability of an area to earthquake hazards. Srinagar has witnessed a population explosion, with the population increasing from 0.25 million in 1961 to 1.5 million in 2011. The city also has a high proportion of female and child residents (59%) and a population density of 4000 people per square kilometer. Migration from rural areas and population growth are the primary drivers of this enhanced population expansion (Nengroo et al., 2018). The city has been under pressure to expand its built-up area in order to cater to the population boom, which has also led to excessive resource depletion, widening wealth and poverty gaps, and detrimental environmental and socioeconomic concerns (Mitsovaa et al., 2010; Kamat and Mahasur, 1997). With the mounting demand for new housing, the quality and condition of houses have received negligible attention. These concerns about accelerated population progression, along with high urbanization, have increased the socio-economic vulnerability of the built environment in Srinagar to earthquakes.

Nengroo, Z. A., Bhat, M. S., and Kuchay, N. A.: Measuring urban sprawl of Srinagar city, Jammu and Kashmir, India, Journal of Urban Management, 6(2), 45-55. https://doi.org/10.1016/j.jum.2017.08.001, 2017.

Mitsova, D., Shuster, W., and Wang, X.: A cellular automata model of land cover change to integrate urban growth with open space conservation, Landscape and urban planning, 99(2), 141-153. https://doi.org/10.1016/j.landurbplan.2010.10.001 2011.

Kamat, S. R., and Mahasur, A. A.: Air pollution: slow poisoning Chennai, The Hindu Survey of Environment, 1997.

**Comment 3:**-In the paper, I think it would be helpful to include a restriction of each of the two models.

**Response:** Thank you for the suggestion. We have added limitations of the two models in the revised manuscript line number 400 to 405 which is reproduced below:

The adopted methodology has a few limitations, much like any other modelling technique. In addition to the inherent flaws in Multi Criteria Decision Analysis (MCDA), there may be some limitations, such as the fact that certain layers become more dominant than others due to the weighting criteria used, which in turn depends upon the decision-makers' perceptions of which vulnerability parameters have the greatest impact on modelling outcomes in vulnerability analysis.

**Comment 4:**- Who, besides the writers, took part in the process of making the expert judgement, and what were their qualifications? Mention these individuals and the expertise that they bring to the table.

**Response:** Thank you for the comment. Though, only the four authors were involved in determining the expert judgement process, viz., Prof. Shakil Ahmad Romshoo, Ph.D., Remote Sensing and GIS; Dr. Irfan Rashid, Ph.D., Environmental Sciences; Dr. Rakesh Chandra, Ph.D., Geology; and Midhat Fayaz, M.Sc. (Geoinformatics), but a large body of literature was also consulted that informed the expert judgement process. A mention of the same has been made in the revised manuscript at line number 336 to 340.

**Comment 5:**- There is a lack of references to certain procedures, such as those pertaining to proximity, closeness, and separation.

**Response:** Thanks for the comment. We have added references for all the approaches used in paper in the revised manuscript.

**Comment 6:**-Both in the introduction and in the end, you made a reference to SDG-11. In the conclusion, however, you should go into more detail on how the findings of this study will contribute to the achievement of SDG-11.

**Response:** Thank you for the suggestion. We have added more details about the SDG-11 in the revised manuscript from line number 619 to 626 under Conclusion section.

The current study is in accordance with the 2030 Agenda for Sustainable Development Goals, which recognises and reiterates the urgent need to lower the risk of disasters. The study will help to reduce the exposure and vulnerability of people to disasters and build resilient infrastructure. The findings of this study will support sensible urban planning, which calls for the construction of resilient infrastructure to reduce vulnerability to natural disasters, as well as sustainable development in line with SDG 11 and SDG 9, which demand manageable densities, user-friendly public spaces, and mixed-use urban development.

**Comment 7:**- There are several typos in the paper which need to be corrected.

**Response:** Thank you very much for this suggestion. We have now rectified all the typos and grammatical errors in the revised manuscript.

**REFREE COMMENTS # 2**

**General comment:** Midhat Fayaz et al. present a vulnerability assessment of the buildings in Srinagar city in the event of earthquake. The authors consider 69 municipal wards within Srinagar urban area. The analysis is based on a building inventory having details of the nature and structure of buildings. The authors perform a ground-based survey to validate the inventory which is commendable. The manuscript offers some insights into differential vulnerabilities of buildings within the urban area. Though there is not any visible flaw in the analysis and assessment some fundamental aspects of earthquake science is missing in the manuscript. Therefore, major revisions as following are required for this manuscript to be considered for publication.

**Response:** Many thanks for commending our work. Authors express their gratitude to the Reviewer for the careful assessment of our work and for valuable suggestions and comments, the incorporation of which have improved the quality of the revised manuscript. We agree with the comment that some of the earthquake related parameters are not included in this study. Below is the point-by-point response to the comments/suggestions.

**Comment 1:-**The authors list major earthquake events in Srinagar city/Kashmir. However, they do not explain how the relate the earthquake related parameters i.e., epicenter with the vulnerability assessments.

**Response:** Thank you for the comment. Agreed that the location of earthquake epicentre indicates the presence of geological structures (faults) in a particular area (Sana, 2018). The available records of historical and instrumental earthquake events (Table 1) in the study area indicate a high probability of earthquake events in the Srinagar city in the future. Dar et al., 2019 have shown that the River Jhelum, running through Srinagar city itself flows along or parallel at many places to a lineament or fault known as Jhelum fault in the Kashmir Valley. Besides high tectonic activity and the lithology (mostly unconsolidated sediments) of the area, makes the area vulnerable to earthquakes.

Therefore, it is believed that in light of the high vulnerability and occurrences of past earthquakes with epicentre in and around Srinagar makes the earthquake vulnerability assessment of study area an important exercise irrespective of the exact location of the epicentre. We assumed that the entire city is vulnerable because of the presence of the Jhelum fault in the midst of the city.

We have modified the study area map in the revised manuscript showing the nearest faults/lineament details (see fig. 1) and made a mention of the above facts in the revised manuscript at line number 124-138.

Sana, H.: Seismic microzonation of Srinagar city, Jammu and Kashmir, Soil Dynamics and Earthquake Engineering, 115, 578-588, https://doi.org/10.1016/j.soildyn.2018.09.028, 2018.

Dar, R. A., Mir, S. A., and Romshoo, S. A.: Influence of geomorphic and anthropogenic activities on channel morphology of River Jhelum in Kashmir Valley, NW Himalayas, Quaternary International, 507, 333-341, https://doi.org/10.1016/j.quaint.2018.12.014, 2019.

[Figure]

Fig.1: Location of the study area. Here MBT stands for Main Boundary Thrust, MCT stands for Main Central Thrust, BF stands for Balapur Fault.

**Comment 2:**- Nowhere it is mentioned how does the geology, lithology and faults are considered to zonate the likelihood of earthquake events within the city. It would have been helpful if the authors have at least considered an earthquake hazard zonation map in their analysis. Without these it is surprising that how the authors assess the earthquake vulnerability of buildings. The present manuscript gives an impression of vulnerability of buildings alone but not for earthquake events.

**Response:** Thanks for the comment. The present study was carried out to look into the vulnerability of built-up environment at a high spatial resolution at building footprint scale ward-wise in the Srinagar city. Geology, lithology, and lineaments faults were taken into account during data analysis. However, keeping in view the high occurrences of the past earthquakes with epicentre in and around Srinagar irrespective of the exact location of the epicentre and distribution of other geological/geomorphic and soil parameters, the entire city wards are equally vulnerable to earthquakes in the eventuality of an earthquake. The River Jhelum, running through the Srinagar city, itself flows along or parallel at many places to a lineament or fault known as Jhelum fault in the Kashmir Valley. Besides because of the high tectonic activity and the lithology (mostly unconsolidated sediments) of the area, it was found that there is very little difference in earthquake vulnerability between various wards because of the similar tectonic, lithologic and geomorphic set up of the city wards; therefore all of these parameters were kept constant. All of Srinagar's wards are situated on consolidated alluvium, or Karewas, which share similar characteristics in terms of how they react to earthquakes. Panjal volcanics are located in a few inconspicuous places, however these are hills that have no habitation, making lithology the least influential parameter in this study. Additionally, there is a very high earthquake risk in each of Srinagar's wards (Sana, 2018; Yousuf and Bukhari, 2020) (see Fig.2). Since the vulnerability at the ward level is the primary focus of the current study, all of the wards were treated as having an equal risk from seismic activity. Please see figure 2 and Figure 3 below for details about the distribution of earthquake related parameter and earthquake hazard map.

Accordingly, we have made a mention of same in the revised manuscript from line number 124-138 under Introduction Section.

Sana, H.: Seismic microzonation of Srinagar city, Jammu and Kashmir, Soil Dynamics and Earthquake Engineering, 115, 578-588, https://doi.org/10.1016/j.soildyn.2018.09.028, 2018.

Yousuf, M., Bukhari, S. K., Bhat, G. R., and Ali, A.: Understanding and managing earthquake hazard visa viz disaster mitigation strategies in Kashmir valley, NW Himalaya, Progress in Disaster Science, 5, 100064, https://doi.org/10.1016/j.pdisas.2020.100064, 2020.

[Figure]

Fig.2: lithology map modified after Thakur and Rawat, 1992.

[Figure]

Fig.3: Seismic Hazard Index map modified after Sana, 2018.

**Comment 3:-** In addition, buildings are susceptible to failures due to ground failure during earthquakes i.e., liquefaction and other soil-structure related damages. Nowhere in this study these aspects have been mentioned or considered.

**Response:** Thanks for the comment. In a similar fashion like that of the lithology, soils and lineaments, the Liquefaction Potential Index (LPI) of the Srinagar shows very least variability from one ward to another. Please see Figure 4 below for the Liquefaction Potential Index of

the Srinagar city (Sana et al., 2016). All the wards of the city fall within LPI of high to very high zone. Thus, like other geological factors, this factor was also kept constant for all the wards of Srinagar under study. It is pertinent to mention here that there are several studies conducted globally that have not included the geological/earthquake parameters in the analysis of earthquake vulnerability assessment of built up environment of cities because of the similar reasons e.g. Srikanth et al., 2010; Ishita and Khandaker 2010; Islam et al., 2013; Alizadeh et al., 2018; Adhikari et al., 2019; Menegon et al., 2019; Fan et al., 2021.

In light of the Reviewer's comments and above explanations, we have made a mention of the above response in the revised manuscript at line number 124-138.

[Figure]

Fig.4: Liquefaction Potential Index map modified after Sana, 2016.

Sana, H., and Nath, S. K. (2016). Liquefaction potential analysis of the Kashmir valley alluvium, NW Himalaya. *Soil Dynamics and Earthquake Engineering*, *85*, 11-18.https://doi.org/10.1016/j.soildyn.2016.03.009

Srikanth, T., Kumar, R. P., Singh, A. P., Rastogi, B. K., and Kumar, S. (2010). Earthquake vulnerability assessment of existing buildings in Gandhidham and Adipur cities Kachchh, Gujarat (India). European Journal of Scientific Research, 41(3), 336-353.

Ishita, R. P., and Khandaker, S. (2010). Application of analytical hierarchical process and GIS in earthquake vulnerability assessment: case study of Ward 37 and 69 in Dhaka City. J Bangladesh Inst Plan ISSN, 2075, 9363.

Islam, M. S., Sultana, N., Bushra, N., Banna, L. N., Tusher, T. R., and Ansary, M. A. (2013). Effects of earthquake on urbanization in Dhaka City. Journal of Environmental Science and Natural Resources, 6(1), 107-112.

Alizadeh, M., Hashim, M., Alizadeh, E., Shahabi, H., Karami, M. R., Beiranvand Pour, A., ... and Zabihi, H. (2018). Multi-criteria decision making (MCDM) model for seismic vulnerability assessment (SVA) of urban residential buildings. ISPRS International Journal of Geo-Information, 7(11), 444.

Adhikari, A., Rao, K. R. M., Gautam, D., and Chaulagain, H. (2019). Seismic vulnerability and retrofitting scheme for low-to-medium rise reinforced concrete buildings in Nepal. Journal of Building Engineering, 21, 186-199.

Menegon, S. J., Tsang, H. H., Lumantarna, E., Lam, N. T. K., Wilson, J. L., and Gad, E. F. (2019). Framework for seismic vulnerability assessment of reinforced concrete buildings in Australia. Australian Journal of Structural Engineering, 20(2), 143-158.

Fan, X., Nie, G., Xia, C., and Zhou, J. (2021). Estimation of pixel-level seismic vulnerability of the building environment based on mid-resolution optical remote sensing images. International Journal of Applied Earth Observation and Geoinformation, 101, 102339.